# Lineage priming and cell type proportioning depends on the interplay between stochastic and deterministic factors

William Salvidge[1], Chris Brimson[1], Nicole Gruenheit[1], Li-Yao Huang[2], Catherine Pears[2], Jason Wolf[3], Chris Thompson[1]*

[1]Centre for Life's Origins and Evolution, Department of Genetics, Evolution and Environment, University College London, London, United Kingdom; [2]Department of Biochemistry, University of Oxford, Oxford, United Kingdom; [3]Milner Centre for Evolution and Department of Life Sciences, University of Bath, Bath, United Kingdom

*For correspondence: christopher.thompson@ucl.ac.uk

Competing interest: The authors declare that no competing interests exist.

## eLife Assessment

This **important** study shows how stochastic and deterministic factors are integrated in *Dictyostelium discoideum* to reliably drive determination of distinct cell types despite exposure to nearly identical environmental conditions. The authors present **convincing** evidence that gene expression variability contributes to the robustness of cell fate decisions, which reveals an unexpected role of stochasticity during cell differentiation.

**Abstract** Isogenic cells can break symmetry and adopt different fates, even when exposed to a seemingly identical environment. This deeply conserved phenomenon allows unicellular organisms to pre-empt dynamically changing environments and is central to the evolution of multicellularity. It is thought that cells are primed towards different lineages by cell-cell variation, although the underlying mechanisms are poorly understood. To address this, we exploit the tractability of the social amoeba *Dictyostelium discoideum*, where cell fate choice also does not depend on spatial cues. We develop and test a model to explain quantitative experimental single-cell observations of probabilistic differentiation. The model suggests that cell cycle position affects lineage choice, as previously shown but that stochastic cell-cell variation also plays a key role. Single cell sequencing reveals genes that exhibit cell type-specific expression or genes that affect fate choice exhibit extensive stochastic cell-cell expression variation. Like lineage priming genes in ESCs, they are associated with H3K4 methylation, which when perturbed affects their expression and disrupt fate choice. We suggest the integration of stochastic and deterministic inputs represents an adaptive mechanism to increase developmental robustness against perturbations that affect deterministic signals.

## Introduction

Symmetry-breaking and cell type differentiation are fundamental features of developmental systems. The mechanisms underlying the spatial control of cell fate choice have been extensively studied (*Slack, 2014*). However, there are many examples of robust cell fate choice in the absence of spatial cues. Examples include embryonic stem cells (ESCs) in culture (*Pauklin and Vallier, 2013*), lineage choice in the mouse blastocyst (*Saiz et al., 2016*; *Yamanaka et al., 2010*) and 'bet hedging' strategies in microbial populations (*Maamar et al., 2007*; *Süel et al., 2007*). In these cases, differentiation is

probabilistic and is thought to depend on cell-cell heterogeneity. Intrinsic variation in cell physiology, redox state, and gene expression arising from differences in cell cycle position, cell size, inheritance of cell contents upon cell division and transcriptional bursting have all been linked to fate choice (*Huh and Paulsson, 2011*; *Raj et al., 2006*). These factors can produce robust cell type proportioning if the probability of cells being in different states is predictable at the cell-population level (if the population size is sufficiently large).

Despite the widespread use of heterogeneity to facilitate developmental decision making, we still have a limited understanding of the underlying mechanisms or how variation is tuned to result in robust proportioning. Work has centred around the identification of factors that 'prime' cells for differentiation before they commit to a particular cell fate and express genes associated with specific cell lineages. For example, genes associated with ESC differentiation exhibit histone modifications associated with both gene activation (e.g. H3K4me3) and repression (e.g. H3K27me3) simultaneously. This has been termed a bivalent state and is thought to result in a poised state that allows developmental genes to be rapidly activated or repressed in response to differentiation cues (*Azuara et al., 2006*; *Bernstein et al., 2006*). H3K4Me3 deposition has also been shown to be sensitive to redox state, thus providing a potential explanation for the link between reactive oxygen species and ESC differentiation (*Ulfig and Jakob, 2024*). It has also been suggested that the level of stochasticity in the expression of lineage-associated genes could influence the number of cells that are in the primed state and thus the number of differentiating cells (*Desai et al., 2021*). Finally, deterministic differences between cells, such as cell cycle position have been shown to influence the response to differentiation cues and lineage choice (*Pauklin and Vallier, 2013*).

The social amoeba *Dictyostelium discoideum* provides a uniquely tractable model system to study the mechanisms underlying lineage priming and the control of robust probabilistic differentiation. In response to starvation *D. discoideum* cells undergo a multicellular developmental programme to build a fruiting body consisting of hardy spore cells and dead stalk cells that aid dispersal from deteriorating environments. There is assumed to be selective pressure to ensure that the 'fittest' or most energy rich cells are chosen as spores, which requires mechanisms that bias differentiation. Furthermore, there is assumed to be strong selective pressure to ensure robust output of 'optimal' cell type proportioning. Excess stalk cell production is costly because it reduces the number of viable spores available for dispersal, while underproduction of stalk can compromise fruiting body architecture, limiting the success of spore dispersal (*Buttery et al., 2009*; *Madgwick et al., 2018*; *Rodrigues and Gardner, 2022*; *Wolf et al., 2015*).

Symmetry breaking and initial cell type differentiation in *D. discoideum* does not depend on spatial cues. Instead, it is generally assumed that *D. discoideum* cells experience similar levels of a stalk-inducing factor, such as DIF-1. DIF-1 is easily diffusible and the rapid movement of cells within an aggregation results in a well-mixed population. Furthermore, levels of DIF-1 are regulated by negative feedback that can buffer perturbations in DIF-1 synthesis (*Insall et al., 1992*). Instead, cell fate choice has been associated with differences in cell cycle position at the time of starvation (*Gomer and Ammann, 1996*; *Gomer and Firtel, 1987*; *Gruenheit et al., 2018*; *Thompson and Kay, 2000*). Cell cycle position affects the threshold of responsiveness to DIF-1 (even though all cells experience the same amount of each signal) (*Chattwood et al., 2013*; *Gruenheit et al., 2018*). Under this scenario, stochastic variation in cell cycle length generates a steady-state distribution of cell cycle positions. Optimal proportioning of stalk and spore cells is presumably reached because cell fate propensity has been evolutionarily tuned through the cell cycle when a sufficiently large number of cells is randomly sampled. It is likely that this provides an adaptive mechanism to select the least fit cells to differentiate as stalk (*Zahavi et al., 2018*). Cells that have just divided are primed to become stalk and are more sensitive to stalk-inducing signals (and less sensitive to spore-inducing signals). Post-mitotic cells have just divided their cytoplasm and are likely to be energy poor compared to cells in G2. This idea is supported by the finding that glucose depletion causes cells to arrest around mitosis and differentiate as stalk cells (*Gruenheit et al., 2018*; *Thompson and Kay, 2000*).

These observations provide insights into the effects of cell cycle position heterogeneity on cell fate at a population level. However, we still have a poor understanding of how single cells interpret heterogeneity to result in probabilistic decision making or how heterogeneity is tuned to result in robust cell type proportions. To understand how cell-cell variation can be exploited to control fate choice and generate robust proportioning in the absence of cell-cell communication, we have used a combination

of mathematical modelling and experimentation. The model is based on observations of probabilistic differentiation in *D. discoideum*. We find that quantitative behaviour can only be explained by a model that not only encompasses deterministic cell cycle effects, but also the effects of stochastic cell-cell variation on the responsiveness of cells to differentiation-inducing signals. Experimental observations, using single-cell RNA-seq, reveal a set of genes that show extensive stochastic gene expression variation at the time of starvation. These genes show many hallmarks of lineage priming genes. They are upregulated as cells undergo a programme of differentiation and exhibit cell-type-specific gene expression. A genome-wide screen reveals that these genes, together with cell cycle-associated genes, are also more likely to be required for cell type differentiation. Stochastically expressed genes also exhibit differential H3K4 methylation. Perturbation of H3K4 methylation preferentially affects the level of their expression (and thus cell-cell variation), as well as cell type proportioning during development. These observations suggest that deterministic variation in cell cycle position acts together with stochastic variation in developmental genes to control probabilistic cell fate choice. Finally, we suggest this system is evolutionarily advantageous because it allows the use of deterministic information on the status or quality of cells (e.g. their energetic state), yet protects against potentially catastrophic effects of large-scale perturbations that cause cells to exhibit inadequate variation in deterministic properties. These studies provide further evidence supporting the key role of stochastic variation in developmental gene expression and cellular decision making.

## Results

### A combination of stochastic and deterministic variation explains lineage priming and fate choice in *D. discoideum*

To understand the mechanisms underlying lineage priming and fate choice, we first reassessed an earlier deterministic model designed to explain cell cycle-dependent fate propensity in *D. discoideum* (*Gruenheit et al., 2018*). The model proposes that a cell cycle-associated factor (CCAF) rapidly accumulates at mitosis and increases stalk propensity (*Gruenheit et al., 2018*). Following mitosis, the level of CCAF decays, resulting in a decreasing propensity of cells to adopt stalk cell fate (and increasing spore propensity) as the cell cycle progresses. This model provides a relatively good fit to observed population level cell fate proportions. However, a deterministic factor governing cell fate would be expected to drive cells in the same state towards the same fate. Instead, experimental observations show cells at the same cell cycle position can adopt different fates (*Gruenheit et al., 2018*). Furthermore, a truly deterministic model would be expected to result in a discrete change from stalk to spore fate at the point in the cell cycle where CCAF level drops below the threshold that results in stalk fate (see *Figure 1* parts A.i and A.ii). Instead, there is a gradual decrease in stalk propensity after mitosis.

To develop a model that generates the sort of probabilistic differentiation observed in *D. discoideum* we first incorporated the deterministic influence of the cell cycle on responsiveness or sensitivity to stalk-inducing factors (CCAF) (*Figure 1Ai*):

$$A_t = A_0 - \beta t \tag{1}$$

where $A_0$ is the starting level of CCAF, $\beta\beta$ is its rate of decay, and $t$ is the amount of time after the end of the previous mitosis. CCAF is measured on the scale of its biological effect rather than in terms of its molecular concentration. In theory, the decay in the level of CCAF could show a range of shapes (e.g. exponential/convex, linear, or parabolic/concave), which depends both on how the underlying factor(s) decay at the molecular level and how molecular concentrations translate into biological effects. We assume linear decay both for simplicity, and because fitting alternative models to experimental data indicate that it is the best fit model (see Supplementary text). Cells with a CCAF level above a threshold value (denoted $R$) adopt stalk cell fate (in response to stalk-inducing factors). As a result, stalk propensity at time $t$ after the end of the previous mitosis ($P_t$) is determined by the proportion of cells with a value of CCAF (given by $A_t$) above $R$. This model would produce the step-like pattern of cell fate through the cell cycle (as described above), where all cells adopt stalk cell fate during the period of the cell cycle where CCAF levels are above the threshold ($A_t \geq R$) and switch to spore cell fate when CCAF levels decay below the threshold ($A_t < R$) (*Figure 1Aii*). This steplike pattern will occur irrespective of how CCAF levels change through the cell cycle (i.e. regardless of

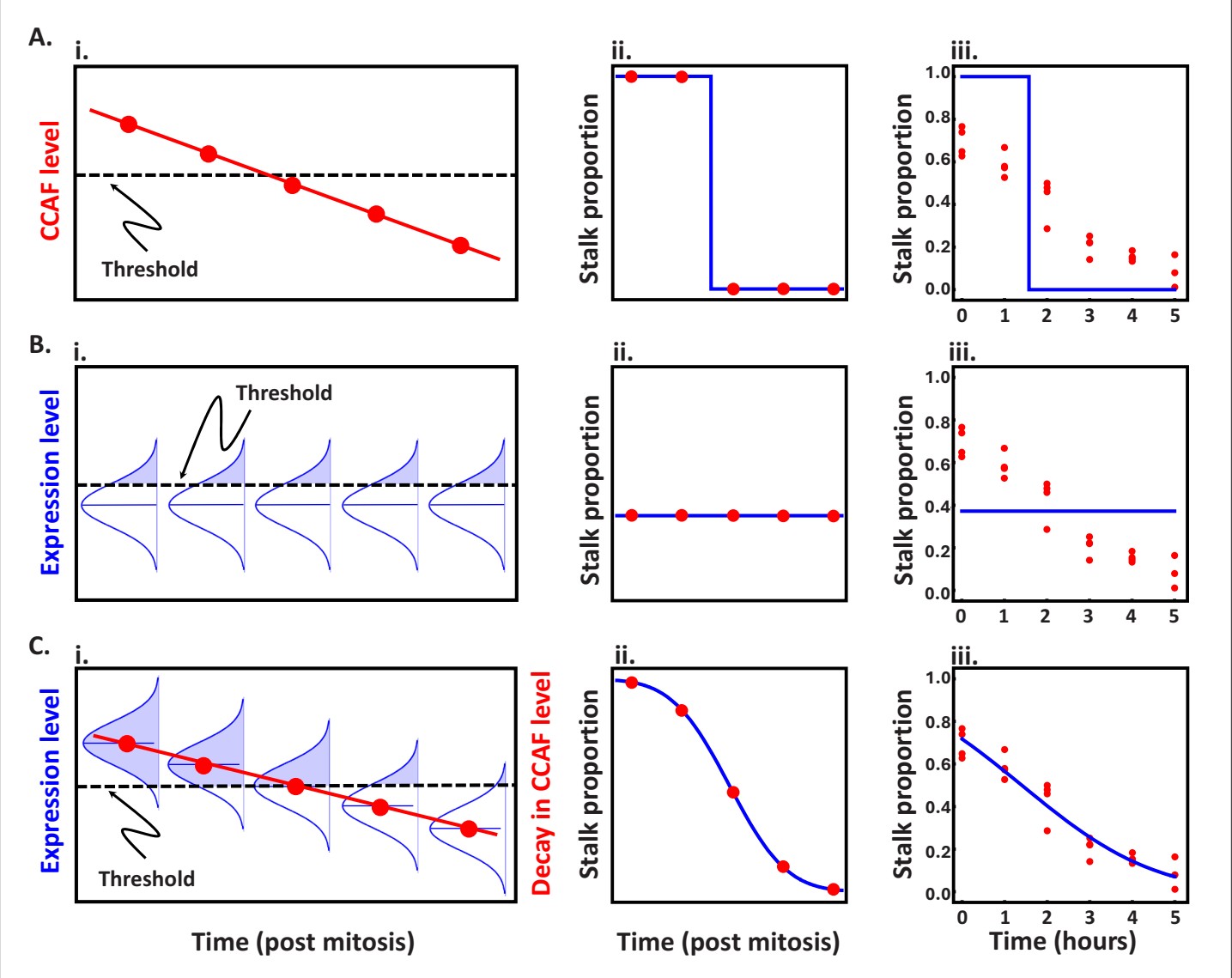

**Figure 1.** Theoretical patterns of stalk proportioning through the cell cycle. Plots illustrate the model structure, theoretical expectations for stalk proportioning ($P_t$) through the cell cycle, and corresponding fits to empirical data (from *Gruenheit et al., 2018*) for scenarios in which cell fate is controlled by either cell cycle position (time post mitosis), noisy gene expression, or a combination of the two. The five red points in the illustrations of the model structure and corresponding theoretical expectations approximately match the five timepoints used in fitting the empirical data. (**A**) Stalk proportioning through the cell cycle is deterministic. (**i**) In this scenario, sensitivity to stalk-inducing factors depends solely on the level of a cell-cycle-associated factor (CCAF), which has a maximal level just after completing mitosis (given by $C_0$) and decays linearly through the cell cycle (at a rate given by $\beta$; see *Equation 1*). (**ii**) When the CCAF level is above a threshold value ($R$), cells adopt stalk fate and when the level drops below the threshold, they switch to spore fate. This results in a stepwise change in cell fate. (**iii**) The best fit to the data shows a clear lack of fit, with the empirical pattern showing neither the predicted all or none pattern of cell fate nor a sharp shift in stalk propensity. (**B**) Stalk proportioning depends solely on noisy gene expression. (**i**) In this scenario, sensitivity to stalk-inducing factors depends on a cell-cycle independent factor (CCIF), whose level is normally distributed (with mean $\mu$ and a standard deviation $\sigma$) because it is determined by the sum across many noisily expressed genes. All cells with a value of CCIF above a threshold value ($R$) adopt stalk fate, while those below the threshold adopt spore fate (see *Equation 2* for the expected stalk proportioning). (**ii**) Because the level of CCIF does not depend on cell cycle, expected stalk proportioning does not change through time. (**iii**) The best fit to the data shows a clear lack of fit, with the empirical pattern showing an obvious shift in stalk propensity through the cell cycle. (**C**) Stalk proportioning depends on the combination of the deterministic cell-cycle dependent change in CCAF and the stochastic input of CCIF from noisy gene expression. (**i**) In this scenario (the stochastic-deterministic model), sensitivity to stalk-inducing factors is determined by the sum of the contribution of CCAF, which decays through the cell cycle, and CCIF, which shows a fixed pattern of variation at all timepoints. The proportion of cells at each timepoint that have a total sensitivity above the threshold value adopt stalk fate, while those below the threshold adopt the spore fate. (**ii**) As CCAF decays through the cell cycle,

*Figure 1 continued on next page*

*Figure 1 continued*

the proportion of cells above the threshold declines, resulting in the non-linear pattern of stalk proportioning seen in the right-hand plot. (**iii**) The best fit to the data shows a very good match between cell fate behaviour and the pattern predicted by the model.

The online version of this article includes the following figure supplement(s) for figure 1:

**Figure supplement 1.** Fit of experimental data to models of fate choice.

---

whether it is linear or nonlinear) since cells in the same cell-cycle state can only be above or below the CCAF threshold that results in stalk fate.

A purely deterministic model clearly does not recapitulate the quantitative shift in cell fate propensity observed experimentally (*Figure 1Aiii*). We next reasoned that, for the system to show a deterministic change through the cell cycle, but result in a probabilistic output (i.e. stalk propensity), it must integrate another source of cell-cell variation that is independent of the cell cycle (along with a deterministic cell-cycle dependent factor, CCAF). We, therefore, hypothesised that there is also a stochastically variable cell-cycle independent factor (CCIF) contributing to sensitivity to stalk-inducing factors (e.g. CCAF accumulation or breakdown). The addition of such intrinsic variability makes cell fate probabilistic by allowing otherwise identical cells to adopt a distribution of possible responsiveness to stalk-inducing factors.

Protein levels can vary widely between cells because it is regulated at multiple levels, including transcription, translation, and stability. The position of the noisiest step in a pathway affects the overall noise dramatically, because each step usually amplifies noise in the previous steps (*Alon, 2007*). Consistent with this idea, theory and single-cell experiments have shown that a major contributor to cell-cell variation is the bursty expression of low-copy mRNAs. We, therefore, hypothesised that this noisiness across cells arises from stochastic expression of a set of genes contributing to CCIF levels. For this, we adapt the logic of the telegraph model of gene expression (*Peccoud and Ycart, 1995*). A given gene is either expressed ($e_i = 1$) or not expressed ($e_i = 0$) in any given time interval being examined, where the probability that it is expressed ($e_i = 1$) is a Bernoulli process with probability $p_i$. Although genes might logically vary in the value of $p_i$, such variability does not impact our results, so we simply consider the average probability of being expressed across all genes: $\bar{p}$. When gene $i$ is expressed, it contributes $x_i$ to the total value of CCIF, making its realised contribution $g_i = e_i x_i$ and expected contribution in any time interval $E[g_i] = p_i x_i = \bar{p} x_i$. We assume that genes vary in their contribution to CCIF according to $x \sim \mathcal{N}(\bar{x}, s^2)$ (i.e. is normally distributed with mean $\bar{x}$ and variance $s^2$). The expected level of CCIF overall all genes at time $t$ is the sum of the $g_i$ values (i.e. the realised level of expression) for the set of $G$ stochastically expressed genes. This expectation gives the mean level of CCIF: $\mu = \sum_{i=1}^{G} g_i = G\bar{p}\bar{x}$, with (following the law of total variance; *Cornell and Benjamin, 1970*) an expected variance of $\sigma^2 = G\bar{p}s^2 + \bar{p}(1 - \bar{p})G\bar{x}^2$. This scenario results in a normal distribution of CCIF levels, CCIF $\sim \mathcal{N}(\mu, \sigma^2)$, which we use to model the properties of the aggregate effect of all stochastically expressed genes on CCIF. Although this derivation implicitly assumes that stochastically expressed genes are independent, this assumption is not strictly required for the distribution of CCIF to be approximately normal. If stochastically expressed genes show clustered co-expression owing to shared regulation, then the sum across these co-expressed blocks is still expected to be approximately normally distributed (as long as there are a reasonably large number of co-expressed clusters) (*Diananda, 1955*; *Hoeffding and Robbins, 1994*; *Rosén, 1967*). In this case, the variance would differ from the form above and instead depend on the properties of the co-expressed blocks (where the blocks of co-expressed genes would simply replace the individual genes in the derivation above).

If we were to only consider the contribution of this stochastic variation, the proportion of cells adopting stalk fate ($P_t$) would be determined by the proportion of the distribution that exceeds the threshold value ($R$) that leads to stalk fate (*Figure 1Bi*), which is given by the complementary cumulative distribution of CCIF values above the threshold $R$:

$$P_t = \frac{1}{2}\left(1 + \mathrm{erf}\left[(\mu - R) / \left(\sigma\sqrt{2}\right)\right]\right), \tag{2}$$

where erf is the Gauss error function, defined as $\mathrm{erf}(z) = (2/\sqrt{\pi})\int_0^z e^{-x^2} dx$. Under this scenario, the proportion of prestalk cells would remain constant throughout the cell cycle (*Figure 1Bii*), which does not follow the observed progressive shift in stalk fate probability after mitosis (*Figure 1Biii*).

We integrate the deterministic influence of cell-cycle state and noisy gene expression by considering that the total level of sensitivity to stalk-inducing factors is the sum of the value of CCAF and CCIF. We achieve this integration by assuming that the mean sensitivity across cells has a constant component (CCIF) given by $\mu$ (*Equation 2*), and a cell-cycle dependent component (CCAF) given by $A_t$ (*Equation 1*). This produces an expression for mean sensitivity at time $t$ ($C_t$) that is analogous to *Equation 1*: $C_t = \mu + A_0 - \beta t = C_0 - \beta t$, where $C_0 = \mu + A_0$ (i.e. is the sum of the mean CCIF value and starting CCAF value). Note that the change in average sensitivity through the cell cycle depends solely on changes in CCAF levels ($\beta t$), but the change in the average stalk propensity does not change linearly through the cell cycle because it depends on the proportion of cells with a value above $R$ (see *Figure 1Ci*). Using this expression, we can rewrite the overall stalk propensity ($P_t$) at time $t$ by replacing $\mu$ in *Equation 2* with $C_t$ (see *Figure 1Ci*):

$$P_t = \frac{1}{2}\left(1 + \mathrm{erf}\left[\frac{C_0 - \beta t - R}{\sigma\sqrt{2}}\right]\right) \tag{3}$$

(which represents the integral over the distribution of sensitivity values above the value $R$ at time t). This 'stochastic-deterministic' model generates a non-linear change in stalk propensity through time (*Figure 1Cii*) that matches the probabilistic nature of cell stalk fate after mitosis (*Figure 1Ciii*). The model is also flexible in that, at its limits, it can produce anywhere from the step function (*Figure 1A*) expected for a purely deterministic process (i.e. as $\sigma \to 0$) and the constant proportioning (*Figure 1B*) expected for a purely stochastic process (i.e. as $\beta \to 0$) (see Appendix 1). To facilitate fitting of the model in *Equation 3* to experimental data, we can reduce the parameter space by combining the three constants ($C_0$, $R$, and $\sigma$) into a single term, denoted $C_0^*$, which represents a sort of reference point for the model. This rescaling is achieved by assuming that the parameters are all measured in standard deviation units and that sensitivity is measured as a distance from the threshold that leads to stalk cell fate ($R$), such that $C_0^* = C_t - R$. This results in a two-parameter version of *Equation 3*, $\frac{1}{2}(1 + \mathrm{erf}[(C_0^* - \beta t)/\sqrt{2}])$, that can be fitted to the stalk propensity data (see below) to estimate the two parameters ($C_0^*$ and $\beta$), where the parameter estimates are in standard deviation units (meaning $C_0^*$ represents how far mean sensitivity is from the threshold, measured as a Z-score). To test this model empirically, we started by fitting the model to existing data on stalk propensity through the cell cycle (see *Supplementary file 1* for the original raw data and adjusted values used in the analysis). These data come from an experiment (*Gruenheit et al., 2018*) where time-lapse microscopy was used to group cells according to their cell cycle position at the time of starvation. The fate of each cell was monitored using live cell reporter genes. For our analysis, we used the measures of relative stalk propensity over 6 hr (after which cells stochastically re-enter mitosis). We find that the stochastic-deterministic model provides an excellent fit to the data (adjusted $R$-squared=0.92, *AIC* = −56.37), yielding estimates of $C_0^* = 0.57$ and $\beta = 0.41$ (*Figure 1Ciii*) that correspond to an expected steady-state stalk propensity of 0.35 and an expected starting propensity of 0.72. Moreover, the stochastic deterministic model provides a significantly better fit to the data than the previously proposed model for exponential decay of CCAF (*Gruenheit et al., 2018*) (it is 98 times more likely; see Appendix 1 and *Figure 1—figure supplement 1*). We also evaluated the support for our model assumptions by comparing the fit of other models derived using alternative assumptions. We evaluated the assumption of linear decay in CCAF by comparison to models with exponential, quadratic, and cubic decay functions (which together capture a broad range of possible shapes), and the assumption of Gaussian variation in CCIF by fitting a model based on gamma-distributed variation (which can capture a diversity of distributions, including ones approximating normality). These model comparisons all support our model assumptions (see Appendix 1 for the methods and results of these comparisons).

## Stochastic gene expression variation is extensive in growing cells

The stochastic-deterministic model suggests that cell fate choice in *D. discoideum* should not only depend on deterministic cell cycle-dependent cell-cell variation, but also stochastic effects on the expression of genes associated with fate choice. Cell cycle position-dependent gene expression variation has been observed in single-cell RNA-seq (scRNA-seq) data from *D. discoideum* cells isolated prior to starvation and exposure to differentiation-inducing signals (*Gruenheit et al., 2018*). Because these data are from a relatively small number of cells, but sequenced to high depth, we reasoned they could also be used to identify stochastic expression variation. We first determined the coefficient of

variation ($CV^2$) of expression for all genes. As expected, this tends to decrease as average expression level increases (*Figure 2—figure supplement 1*). When this trend is accounted for, genes with greater cell-cell variability than expected for their level of expression (FDR <0.05) were identified (*Figure 2A*). To remove those genes where variation can be explained by cell cycle position, we used Area Under the Receiver Operating Characteristic Curve (AUROC) to identify those genes where expression was associated with previously defined cells in early G2, late G2, or M/S phase (*Gruenheit et al., 2018* and *Figure 2—figure supplement 1*). A conservative AUROC threshold (>0.65) allowed even those genes that show a weak association with cell cycle position to be identified. When these were removed, many genes remain that exhibit variation that cannot be explained by differences in cell cycle position (*Figure 2B*). Unlike cell cycle genes, principal component analysis does not result in cell groupings (*Figure 2C*). This variation is, therefore, consistent with stochastic influences on gene expression rather than a consequence of hitherto unknown extrinsic cues. This approach was extended to determine whether cell cycle-associated genes are also influenced by stochastic effects. For this, the $CV^2$ of each gene was recalculated within groups of cells from each of the different cell cycle stages. Most cell cycle dependent genes were found to exhibit greater within group variation than expected (*Figure 2D*). This is not due to the low level of expression at specific cell cycle stages, as variation is higher at all stages, including when they are maximally expressed (*Figure 2—figure supplement 2*). These results thus reveal that stochastic effects on gene expression variation are widespread in growing cells.

## Stochastically expressed genes are associated with cell fate determination

In *D. discoideum*, growing cells undergo a development cycle that begins with the aggregation of thousands of cells and ends with the formation of a fruiting body consisting of terminally differentiated stalk and spore cells. To test whether genes that exhibit variability in their expression are associated with cell fate choice, we first compared the timing of expression of stochastically expressed genes, cell cycle-associated genes, and non-variable genes. The average expression of each gene was compared during growth and development to generate a developmental index (where 0 is exclusive to growth and 1 is exclusive to development) (*de Oliveira et al., 2019*). Stochastic genes were greatly enriched ($p \leq 0.001$, binomial test) for developmental genes (index $\geq 0.9$), whereas cell cycle and non-variable genes showed no enrichment compared to the genome-wide expectation ($p > 0.5$, binomial test) (*Figure 3A*). Next, we tested whether any of these groups of genes were associated with stalk or spore cell fate. Precursors of stalk and spore cells can be identified in the multicellular slug. First, RNA-seq data from prestalk and prespore cells was analysed to identify genes that exhibit cell-type-specific gene expression (where 0 is exclusive to prespore cells and 1 is exclusive to prestalk cells). Again, stochastic genes were significantly enriched in cell-type-specific genes ($p \leq 0.001$), with both prestalk ($p \leq 0.001$, binomial test) and prespore genes ($p = 0.013$) contributing to this enrichment (*Figure 3B*). Finally, we determined whether stochastic genes are more likely to be required for fate choice. We performed an unbiased large scale REMI-seq forward genetic screen (*Gruenheit et al., 2021*) to identify genes required for stalk cell differentiation. REMI-seq technology permits the abundance of thousands of mutants to be simultaneously quantified before or after a selection regime, such as selection imposed by the ability to undergo prestalk cell differentiation. The REMI-seq library was plated at low cell density and treated with cAMP to induce competence to differentiate followed by treatment with DIF-1 to induce prestalk cell differentiation (*Figure 3C*). Prestalk cells terminally differentiate as dead stalk cells after prolonged DIF-1 incubation, and thus surviving mutants with defects in prestalk cell differentiation can be enriched. After 2 and 6 rounds of growth and selection, gDNA was prepared from each biological replicate for sequencing and quantitative analysis. In order to ensure that enrichment was due to a failure to respond to DIF-1, rather than increased growth rate, we compared these mutants to a control selection in which cells were simply taken through an equivalent number of generations of growth (*Gruenheit et al., 2021*). An additional control screen was performed to identify mutants that are incompetent to differentiate at all (as either stalk or spore) because they fail to respond to cAMP. Cells were incubated in the presence of the cAMP analogue, 8-Br-cAMP, which triggers spore cell differentiation. Cells that cannot differentiate as spores were killed by detergent treatment, thus reducing their frequency in the population. After these mutants were removed, 244 mutants remained that have likely been

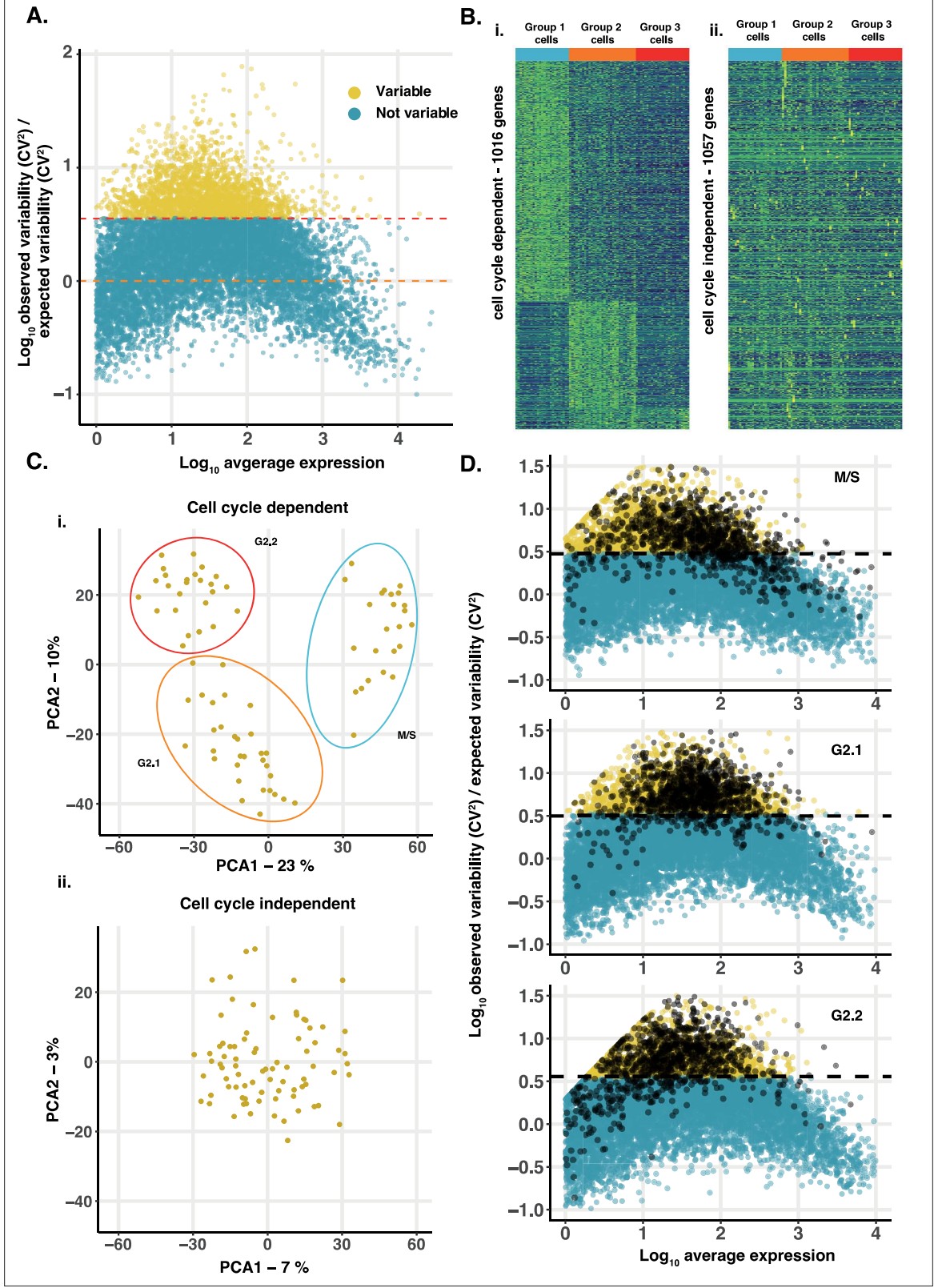

**Figure 2.** Identification of gene expression variation. (**A**) Gene expression variability in single cells. Plot shows $\log_{10}$-transformed relative gene expression variability (observed $CV^2$/expected $CV^2$) versus $\log_{10}$-transformed average gene expression level for all genes in 81 single cells. 2073 genes that show significantly greater variation (FDR <0.05; red dashed line) than expected (black dashed line) were identified (yellow dots). (**B**) Variable genes can be divided into cell cycle associated and cell cycle-independent genes. Variable genes were defined as cell cycle-associated (1016 genes) or cell

*Figure 2 continued on next page*

*Figure 2 continued*

cycle-independent (1057 genes). (**i**) Hierarchical clustering reveals cell cycle-associated genes show a strong cell cycle-associated gene expression pattern blue (S/M phase, 669 genes), orange (G2.1 phase, 247 genes) and red (G2.2 phase, 100 genes). (**ii**) Cell cycle independent genes show little or no pattern with respect to the cell cycle groups. Cell order is the same in both plots. (**C**) Non-cell cycle-associated genes do not drive cell groupings. (**i**) Plotting PC1 against PC2 from a PCA of expression variation in cell cycle-associated variable genes identifies three cell groups (blue, orange, red circles). (**ii**) PCA of non-cell-cycle variable genes does not identify any grouping of cells. (**D**) Cell cycle dependent genes also exhibit extensive stochastic cell-cell variation. The $CV^2$ was calculated for all genes within each group of cells at different cell cycle stages and compared to the expected value based on average gene expression level. Genes with more variation than expected (adjusted *p*-value <0.05, above dashed line) are shown in yellow. The 1016 cell cycle-associated genes (black) are generally more variable than expected at each cell cycle stage.

The online version of this article includes the following figure supplement(s) for figure 2:

**Figure supplement 1.** Cell cycle dependent genes defined through AUROC analysis.

**Figure supplement 2.** Markers of different cell cycle phases also exhibit more variation within cells at the same cell cycle positions.

enriched due to defects in stalk cell differentiation (*Figure 3D*, *Supplementary file 1*). A subset of these mutants was recreated in the parental strain and tested in stalk cell induction assays and most mutants (8/9) exhibited defects in stalk cell differentiation (*Figure 3E*), illustrating the quantitative success of the REMI-seq approach. This allowed us to test whether genes required for differentiation are more likely to exhibit cell-to-cell gene expression variation. REMI mutants were assigned to 199 genes with intragenic insertions, or mutations where the REMI insertion lay within upstream promoter sequences (within 500 bp of the transcription start site). These genes were significantly enriched for genes with variable expression (*p*≤0.001, binomial test) (*Figure 3F*). This is due to both cell cycle-associated and stochastically expressed genes, as the relative number of genes identified in each class does not vary significantly from expected (chi-square, *p*=0.77). Genes that affect fate choice also exhibit greater variability than expected when the $CV^2$ was normalised to their expression (***$p$≤0.001, t-test) (*Figure 3F*). These results suggest gene expression variation is a feature of genes associated with fate choice and cell type proportioning.

## Stochastically expressed developmental genes exhibit differential patterns of H3K4 methylation

Fate choice in *D. discoideum* shares features with lineage priming in embryonic stem cells, including extensive cell-cell variation of genes associated with differentiation (*Chang et al., 2008*). In embryonic stem cells, these genes are also associated with specific patterns of epigenetic marks, including the co-occurrence of H3K4me3 and H3K27me3. The role of these modifications is not fully understood due to the difficulty with which epigenetic marks can be altered at a genome-wide scale in higher organisms. However, recent studies have linked histone modifications to the control of transcriptional burst frequency (*Weinberger et al., 2012*; *Wu et al., 2017*), which will in turn affect the level of cell-cell variation in transcription. The apparent absence of polycomb-like proteins in *D. discoideum* suggests H3K27me3 modification is unlikely to play any role (*Kaller et al., 2006*). However, H3K4 mono, di, or tri-methylation (H3K4Me1-3), which is dependent on Set1/COMPASS, is present (*Chubb et al., 2006*). H3K9/K14 acetylation is present, which is consistent with the idea that H3K4me3 targets the Gcn5 H3K9/K14 histone acetyl-transferase to specific loci (*Huang et al., 2021*). We, therefore, tested whether genes that exhibit cell-cell variation in expression in *D. discoideum* also show hall-marks of differential regulation by H3K4 methylation. Analysis of ChIP-seq data (*Wang et al., 2021*) revealed that H3K4 methylation exhibits a characteristic gene expression level-dependent pattern around the transcription start site and gene body (*Barski et al., 2007*; *Soares et al., 2017*; *Figure 4A*). To compare patterns in variable and non-variable genes, it was thus necessary to divide genes into bins with similar gene expression levels. Ten random samples of non-variable genes with the same distribution of expression levels as those seen in variable genes were used to compare the profile and number of genes with H3K4 methylation (*Figure 4—figure supplement 1*). This revealed variably expressed genes exhibit different profiles around the gene promoter and gene body (*Figure 4B*). In addition, when these were divided into stochastic and cell cycle genes, both groups were enriched for H3K4Me1 (binomial test *p*<0.001, *Figure 4C*), whilst stochastic genes were depleted for H3K4Me3 (binomial test *p*<0.001, *Figure 4C*).

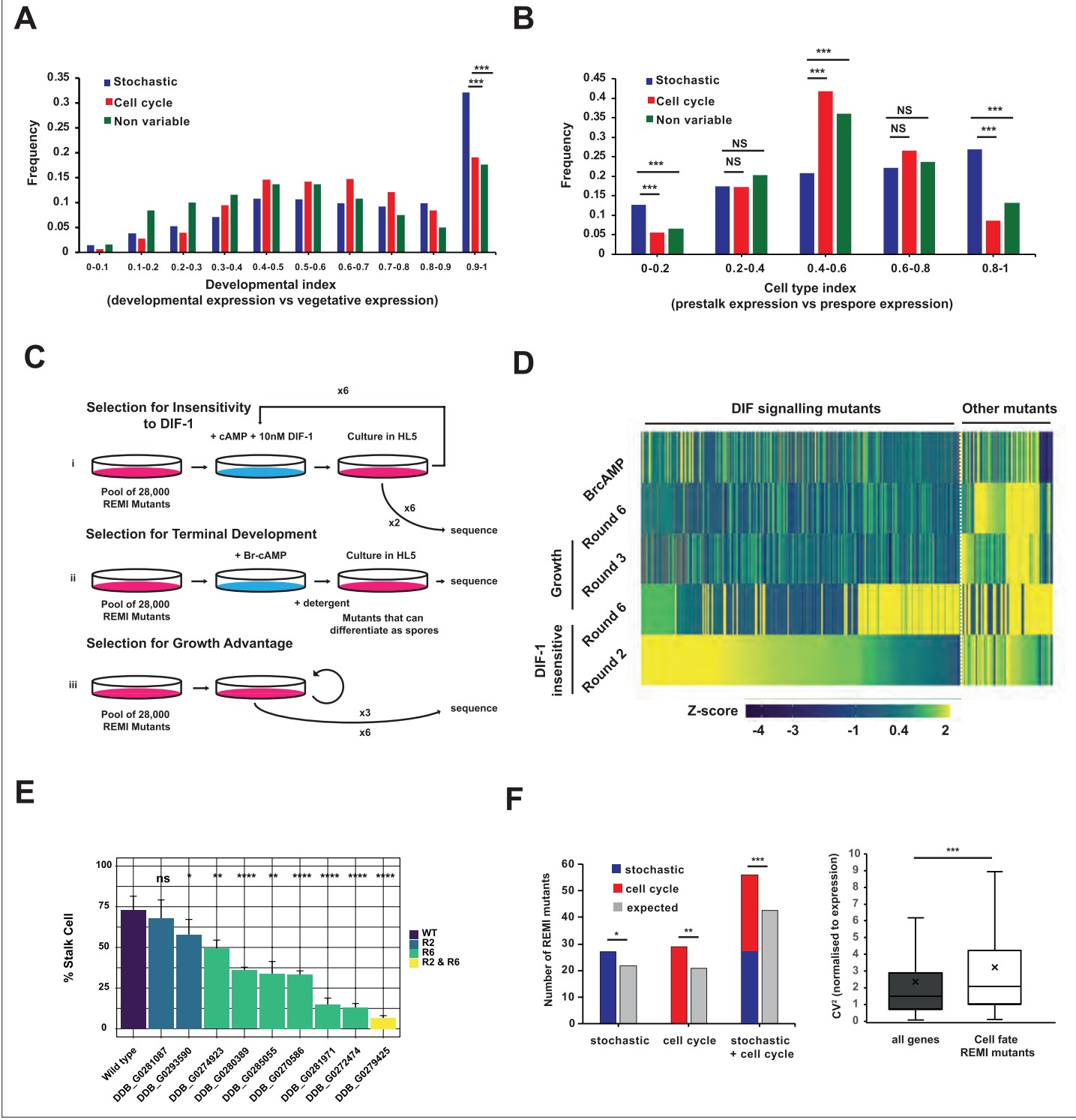

**Figure 3.** Variably expressed genes are important for prestalk cell differentiation. (**A**) Developmental index of different groups of genes. Average gene expression during developmental stages was compared to average expression during growth + average expression during development. This developmental index ranges from zero to one. Zero is exclusive to growth and one is exclusive to development (***$p \leq 0.001$). (**B**) Cell type index of different groups of genes. Gene expression in prestalk cells was compared to expression in prestalk + prespore cells. This cell type index ranges from zero to one (zero is exclusive to prespore cells and one is exclusive to prestalk cells). (***$p \leq 0.001$). (**C**) Schematic of screens carried out to identify mutants defective in prestalk cell differentiation. (**i**) Selection for DIF-1 insensitivity. Cells were incubated with cAMP and DIF-1 to induce stalk cell differentiation. Only cells that fail to respond to these signals remain as amoebae and can re-enter growth upon addition of growth medium. gDNA was extracted for sequencing after two or six rounds of selection. (**ii**) Selection for competence to respond to cAMP. Cells were treated with 8-Br-cAMP to

*Figure 3 continued on next page*

*Figure 3 continued*

induce terminal spore cell differentiation, before treatment with detergent to remove cells that failed to respond and remain as amoeba. A single round of this assay followed by sequencing allowed mutants that dropped out (failed to respond to 8-Br-cAMP to be identified). (**iii**) Selection for mutants with growth advantage. Cells were grown for a similar number of generations to that estimated from re-growth after stalk cell induction. (**D**) Identification of mutations that affect stalk cell differentiation. Heatmap of all 315 REMI mutant positions that were enriched in round 2 or 6 of the DIF-1 selection. Mutants enriched in the screen were removed because they increased in frequency after only growth (63 growth mutants) or failed to differentiate in response to Br-cAMP (8 cAMP non-responsive mutants). 244 insertions were classified as DIF signalling mutants that affect stalk cell differentiation. Level of enrichment is shown as a Z-score. (**E**) Validation of mutants identified by REMI-seq. Stalk cell inductions were carried out on independently generated mutants from round 2 or 6. 8/9 mutants produce fewer stalk cells than wild-type (*$p \leq 0.05$, **$p \leq 0.01$, ***$p \leq 0.001$, ****$p \leq 0.0001$). Results are an average of at least three independent experiments. (**F**). REMI mutants that affect development are enriched for variable genes. (**i**) Comparison of expected vs observed numbers of REMI mutants with variable, cell cycle, or stochastic expression ii. Comparison of normalised CV$^2$ of genes associated with REMI mutants that affect development vs the genome-wide average (*$p \leq 0.05$, ** $P \leq 0.01$, *** $P \leq 0.001$).

## H3K4 methylation is required for normal expression of stochastic genes

Variably expressed genes exhibit different patterns of H3K4 methylation compared to non-variable genes. To test whether their expression also depends on H3K4 methylation, we examined a *D. discoideum* Set1 knockout strain in which H3K4 methylation is abolished (**Chubb et al., 2006**; **Figure 5A**). Bulk RNA-seq was performed on wild-type and Set1 mutant cells. Most genes increased in expression, with 459 up-regulated, compared to 132 down-regulated genes in *Set1*$^-$ mutant cells (**Figure 5B**). The affected genes were strongly enriched for variable genes identified by scRNA-seq, and the level of enrichment increases as the stringency with which variable genes are defined is increased (**Figure 5C**). This suggests that Set1-dependent genes are among the most variably expressed genes during growth (**Figure 5C**, **Figure 5—figure supplement 1**). Furthermore, if variable genes are divided into cell cycle and non-cell cycle-associated stochastic genes, Set1-dependent genes are enriched in both groups (binomial test, $p<0.01$) (**Figure 5D**), suggesting a general role in controlling the expression of variably expressed genes associated with fate choice. These findings were validated with reporter genes in which the promoters of representative Set1-dependent variable genes (*GtaU*, *HspF-2*) were used to drive RFP expression (**Figure 5—figure supplement 2**). The *actin15* promoter was used as a control because it exhibits little cell-cell expression variation and is unaffected by disruption of Set1 in RNA-seq. Actin15-RFP expression was similar in wild-type and Set1$^-$ mutant cells (**Figure 5—figure supplement 2**), but both *GtaU* and *HspF-2*-RFP expression increased markedly in most cells (**Figure 5—figure supplement 2**).

Changes in gene expression can be caused by changes in burst frequency or burst size. To better understand the effects of Set1 on gene expression, we adapted the stochastic-deterministic model to examine the properties of expression variation of individual genes instead of their aggregate influence on CCIF levels. For this, the probability that a gene is expressed, $p_i$, can be interpreted as a measure of the burst frequency, since burst frequency will dictate the probability of observing gene $i$ being expressed in interval $t$t. When gene $i$ is expressed, it produces an amount of transcript given by $b_i$, which is a measure of burst size. Cells in the 'off' state are assigned a value of $b_i = 0$. There can be variability in the level of expression of gene $i$ among cells in the 'on' state, denoted $\varepsilon^2$, which can reflect both error variation in experimental estimation and biological noise/variability across cells, such that the distribution among these cells is $\sim N[b_i, \varepsilon^2]$. Hence, the average level of expression for gene $i$ across a set of cells (which includes cells in both the on and off states) is $p_i b_i$, while its variance in expression would be $p_i \varepsilon^2 + p_i(1 - p_i)b_i^2$. Consequently, an increase in burst frequency would be expected to lead to an increase in mean expression. Because both the expected mean and variance in expression levels change as a function of a change in burst frequency, we can measure the impact of a change in burst frequency on expression variability as the squared coefficient of variation (CV$^2$), which has an expected value of $(1 - p_i)/p_i + \varepsilon^2/(p_i b_i^2)$. The value of CV$^2$ would, therefore, be expected to decrease if burst frequency were to increase (i.e. $\Delta p_i > 0$) and vice versa.

To test how Set1 affects gene expression, we examined the effects of Set1 disruption on gene expression in single cells. scRNA-seq data was generated from wild-type and *set1*$^-$ mutant cells. The level of gene expression and cell-cell variation (CV$^2$) was determined for genes shown by bulk RNA-seq to be under the control of Set1 and by high depth scRNA-seq to exhibit significant variation in wild-type cells. We find that the average expression level of these genes was affected in *set1*$^-$ mutant cells

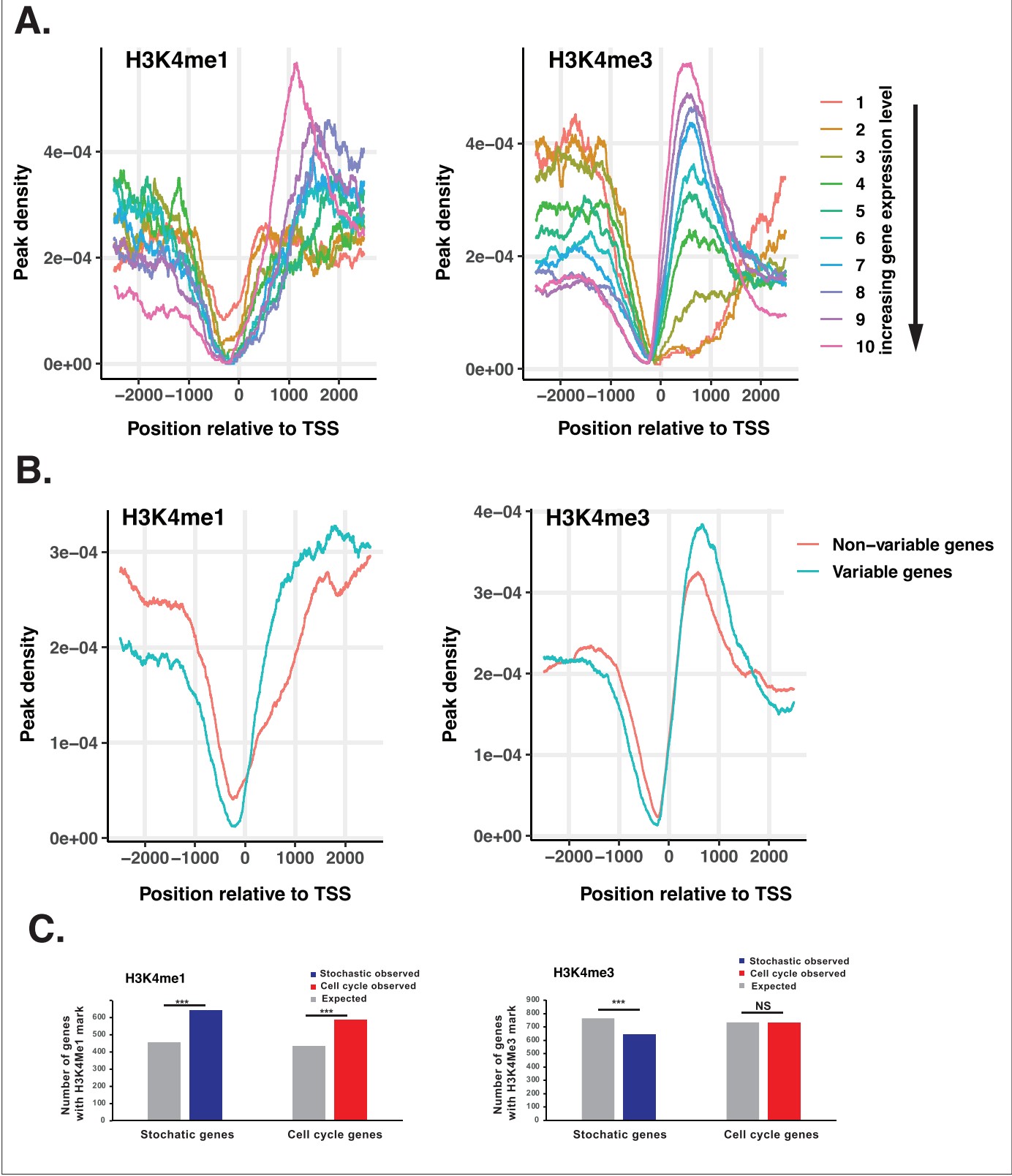

**Figure 4.** Variably expressed genes exhibit differential patterns of histone modifications. (**A**). The H3K4 methylation profile in *D. discoideum*. Genes were assigned to ten bins according to their average expression level in vegetative AX4 cells (Supplementary data). Genes were identified that contain H3K4me1 or H3K4me3 peaks around the TSS (–2500 bp to + 2500 bp) and the distribution of peaks for each bin was calculated. Both H3K4me1 (left) and H3K4me3 (right) exhibit characteristic distributions around the TSS, which correlate with expression level. (**B**). H3K4 methylation patterns differ in

*Figure 4 continued*

variable and non-variable genes. The peak density of variable genes with at least one H3K4me1 or H3K4me3 peak between 2500 bp upstream and 2500 bp downstream of the TSS was plotted (blue line). To control for the effect of expression level on peak density, the distribution of expression levels of variable genes was used to select ten random samples of non-variable genes with the same expression distribution (*Figure 4—figure supplement 1*). The peak density of the ten random samples was averaged (red line). (**C**). Number of genes with H3K4 methylation Comparison of the numbers of observed and expected genes associated with H3K4Me1 and H3K4Me3 marks (***$p \leq 0.001$).

The online version of this article includes the following figure supplement(s) for figure 4:

**Figure supplement 1.** Normalisation of expression levels of variable and non-variable genes for ChIP analyses.

(*Figure 5E*). As expected, based on the model predictions, the level of cell-cell variability measured by $CV^2$ significantly decreased in up-regulated genes (*Figure 5F*), which is consistent with an increased burst frequency. These findings are supported by quantification of *GtaU* and *HspF-2* reporter gene expression levels and cell-cell variability (*Figure 5G and H*). The observed changes in gene expression are thus consistent with the idea that disruption of Set1 affects gene expression by increasing burst frequency and thus the number of expressing cells (which also reduces cell-cell gene expression variation) rather than simply further increasing the level of expression in the subpopulation of cells that already express these genes.

## Loss of Set1 dependent H3K4 methylation affects cell fate choice

Genes associated with cell fate choice exhibit cell-cell variation in gene expression, they are enriched for Set1-dependent histone modifications, and their normal expression is dependent on Set1. Recent studies have also suggested that mutation of Set1 has effects on development and responses to stalk-inducing signals, potentially through its effects on TORC1 activity (*Wang et al., 2024*). We, therefore, further investigated the effects of Set1 disruption on fate choice. Disruption of Set1 did not result in defects in developmental timing, but slugs exhibit a slightly elongated and twisted morphology (*Figure 6—figure supplement 1*). Clear defects were, however, evident in chimeric development between wild-type and *set1*- mutant cells (*Figure 6A*, *Figure 6—figure supplement 2*). *set1*- mutant cells sorted strongly towards the posterior and collar regions in chimeric slugs (*Figure 6A*) and to the upper and lower cup of the fruiting body, which are normally occupied by prestalk (pstB and/or pstO) cell subtypes (*Figure 6—figure supplement 2*). This sorting behaviour suggests H3K4 methylation is required for normal pre-spore cell differentiation, and that mutant cells instead differentiate as pstB and/or pstO cell types. Indeed, when RNA-seq was performed on *set1*- cells separated by FACS after chimeric development, *set1*- cells express pre-stalk genes more strongly, whereas pre-spore genes are poorly expressed (*Figure 6B*). Chimeric development of wild-type and *set1*- strains transformed with prestalkO (ecmO-lacZ), prestalkB (ecmB-lacZ), or prespore (*pspA-RFP*) reporter constructs further confirm these findings. Both *ecmB*-lacZ and *ecmO*-lacZ reporter genes are more strongly expressed in *set1*- cells than wild-type (*Figure 6C*), whilst the pspA-RFP marker gene is only very weakly expressed (*Figure 6C*). We next used RNA-seq transcriptome profiling as it provides a sensitive method to detect defects in developmental gene expression during clonal development. Genes normally expressed in pre-stalk cells were found to be overexpressed, whereas pre-spore genes are poorly expressed (*Figure 6D*). Furthermore, representative pre-stalk lacZ marker genes with a long half-life, are strikingly mis-expressed throughout the slug (*Figure 6E*).

The data from chimeric and clonal development suggest *set1*- mutant cells exhibit an intrinsic defect in pre-spore cell differentiation and tendency to differentiate as pstO and pstB cell types. To determine if this is due to a lack of H3K4 methylation, rather than some hitherto unknown function of Set1, a strain was generated in which another COMPASS complex component, Ash2, was knocked out. Again, this resulted in a severe reduction in the levels of H3K4 methylation (*Figure 6F*). A strain was also generated in which Gcn5, which is required for H3K9/14 acetylation, was disrupted (*Figure 6G*). The effects of H3K4 methylation on gene expression are thought to be, in part, due to its effect on targeting H3K9/14 residues for acetylation. Indeed, disruption of *set1* in *D. discoideum* has previously been shown to decrease H3K9/14 acetylation (*Hsu et al., 2012*). Both *ash2*- and *gcn5*- mutants pheno-copied *set1*- knockout cells in chimeric development with wild-type cells (*Figure 6G*). In addition, chimeric development of a hypomorphic Set1-FLAG knock-in allele (which results in reduced di-methylation and absence of tri-methylation but little effect on mono-methylation *Figure 6—figure supplement 3*) results in a qualitatively weaker sorting phenotype than that seen with *set1*- or *ash2*- mutants

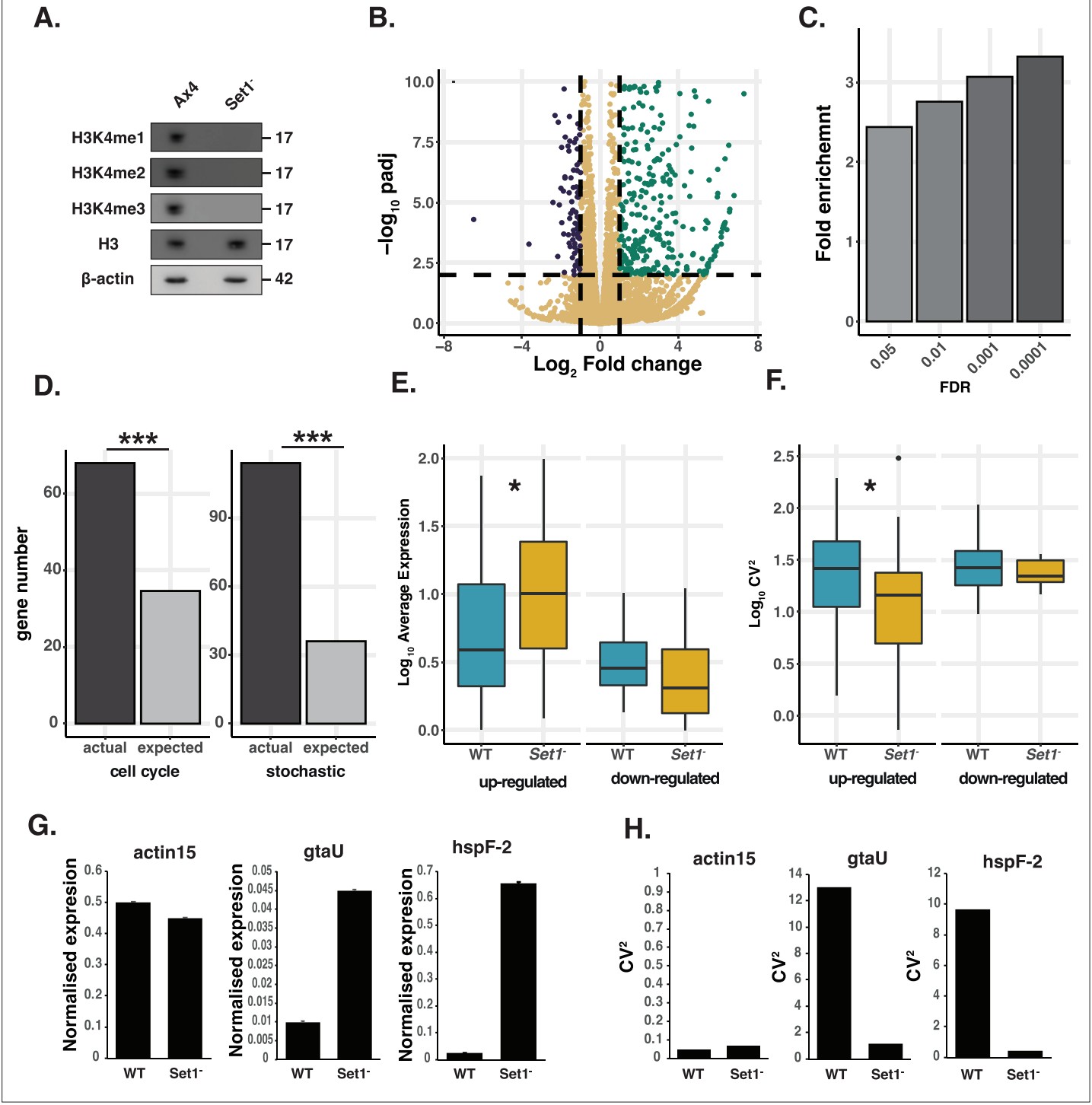

**Figure 5.** Set1 controls cell-cell gene expression variation. (**A**). Set1 is required for H3K4 methylation. Western blot of mono, di, and tri methylation at position H3K4 in wild-type Ax4 and mutant *Set1⁻* cells. Disruption of *Set1* leads to the complete loss of all three methylation marks. Histone 3 levels are unaffected. (**B**). Deletion of *Set1* leads to specific changes in gene expression. Volcano plot showing differential gene expression between *Set1-* mutant and wild-type cells. Up-regulated (green) and down-regulated (blue) genes are highlighted based on cut-offs of two-fold change and an adjusted p-value of <0.01. (**C**). Set1-dependent genes are enriched for variable genes. Quantification of enrichment of Set- dependent genes in variable gene populations. As the threshold for the definition of a variable genes increases, the enrichment for *Set1* repressed genes also increases. (**D**). Set1-dependent variable genes are enriched for cell cycle-associated and cell cycle independent stochastically variable genes. Plot shows the number of cell cycle-associated and cell cycle independent stochastically variable genes that are affected by Set1 disruption. This is compared to a random sample of the same number of genes from the genome. *** binomial test, *p*<0.01. (**E**). Variable genes show increased expression in *Set1-* single cells. Expression

*Figure 5 continued on next page*

*Figure 5 continued*

level was calculated from 310 wild-type and 310 *Set1*⁻ cells. Variable genes (FDR <0.05) were defined as those up- or down-regulated in the bulk *Set1* mutant sequencing. Upregulated genes show a statistically significant increase (*p*<0.01) in expression. Downregulated genes show a small non-significant decrease in expression (**F**). Variable genes show decreased expression variability in *Set1*⁻ single cells. CV² was calculated from 310 wild-type and 310 *Set1*⁻ cells. Variable genes (FDR <0.05) were defined as those up- or down-regulated in the bulk *Set1* mutant sequencing. Up-regulated genes display a significant decrease in CV² in the Set1 mutant compared to WT cells (*P*<0.05). Downregulated genes show a small non-significant decrease in variability. (**G**) *GtaU* and *HspF-2* show increased expression in *Set1*⁻ single cells. FACS quantification of *GtaU* and *HspF-2* reporter gene expression levels normalised to actin15-GFP. (**H**) *GtaU* and *HspF-2* show decreased cell-cell variability in *Set1*⁻ single cells. Cell-cell variability in normalised expression (CV²) from FACS analysis of *GtaU* and *HspF-2* reporter genes.

The online version of this article includes the following source data and figure supplement(s) for figure 5:

**Source data 1.** PDF file containing original western blots for **Figure 5A**, indicating the relevant bands and treatments.

**Source data 2.** Original files for western blot analysis displayed in **Figure 5A**.

**Figure supplement 1.** Set1-dependent genes are more variable than expected.

**Figure supplement 2.** Dual reporter gene analysis of gene expression variation.

---

(**Figure 6—figure supplement 3**). Together, these results support the idea that H3K4 di and tri methylation is required for normal pre-spore cell differentiation.

## Cell cycle position and gene expression variation interact to control cell type proportioning

Set1 disruption predominantly affects the expression of stochastically expressed developmental genes and results in defects in cell fate choice during development. This is consistent with predictions of the stochastic-deterministic model at the single-cell level. A perturbation to the regulation of stochastic expression that increases the probability that genes will be expressed (i.e. increases $\bar{p}$) will necessarily lead to an increase in the average level of CCIF (since the mean $\mu = G\bar{p}\bar{x}$, and such a perturbation would increase $\bar{p}$), and thus will increase stalk cell differentiation. This, however, assumes that the regulation of CCAF and CCIF are independent and there are no effects on the cell cycle or the level of CCAF. To test this, we analysed the effects of perturbing cell cycle-associated gene expression variation on stochastic variation, and vice versa. Cell cycle progression was blocked through cold shock, which results in widespread changes in cell cycle-associated gene expression (**Strasser et al., 2012**). However, this did not affect the expression of representative Set1-dependent reporter genes that are not associated with the cell cycle (**Figure 7A and B**). Next, we perturbed the level of expression of variably expressed genes through Set1 disruption. scRNA-seq reveals *set1*⁻ cells still exhibit cell cycle-associated differential gene expression with the number of M/S and G2 phase cells unaffected (**Figure 7C**, **Figure 7—figure supplement 1**). Furthermore, cell cycle progression and timing of cell division are unaffected by Set1 disruption (**Figure 7D**). These data thus suggest Set1-dependent variable genes and cell cycle-dependent variation are independently controlled.

The stochastic deterministic model predicts an increase in CCIF should increase the likelihood of each single cell responding to stalk-inducing signals (such as the stalk inducer DIF-1). To test this idea, DIF responsiveness was quantitatively compared at the single cell level in wild-type and *set1*⁻ mutant strains in which a GFP reporter gene had been knocked into the DIF-sensitive *ecmA* or *ecmB* prestalk genes (**Figure 8—figure supplement 1**). The number of *ecmB* GFP-positive cells was significantly higher in *set1*⁻ mutant cells at all DIF-1 concentrations tested, and the number of *ecmA*-positive cells was higher at the lowest concentrations (**Figure 8A and B**). However, the level of GFP expression in each responding cell did not significantly change (**Figure 8A and B**). This suggests that the number of cells that are in a DIF responsive state is dependent on the level of Set1-dependent stochastic gene expression (CCIF), as well as the position in the cell cycle (CCAF). We next analysed the relationship between cell cycle phase and cell fate choice in *set1*⁻ mutant type cells. Cells were filmed growing at low density for 12–14 hr, which allowed each cell to be tracked for at least one cell division. The growth medium was then removed, and cell type differentiation induced. Cell tracking was continued, and the final fate of each cell (expression of cell-type-specific GFP reporter gene) was compared to the cell cycle position (timing of the last division) when differentiation was induced. Consistent with expectation from the stochastic-deterministic model (where expected stalk propensity, $G\bar{p}\bar{x}$, increases will burst frequency), *set1*⁻ mutant cells exhibit a higher probability of adopting the pre-stalk cell fate

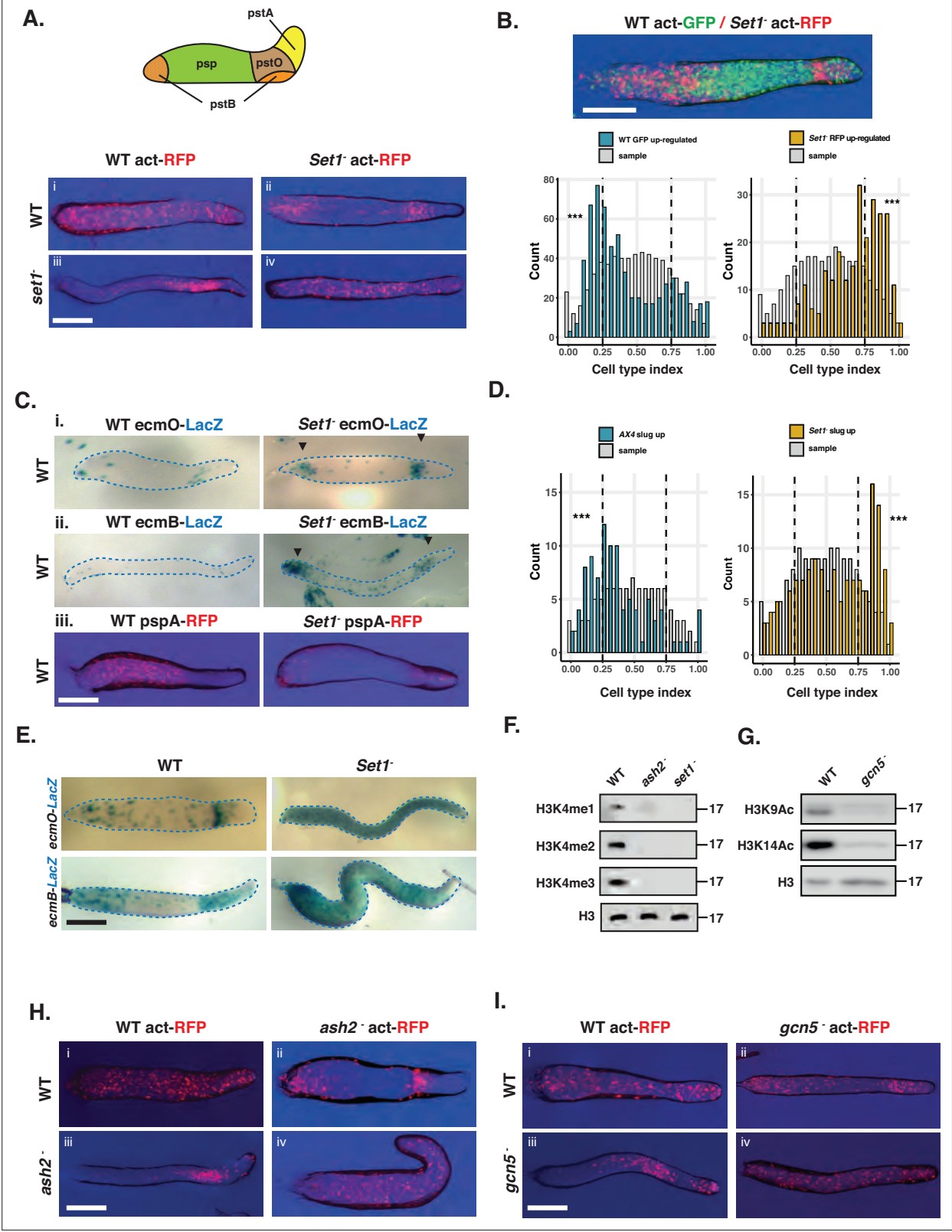

**Figure 6.** Set1 is required for cell fate choice. (**A**) *set1⁻* mutant cells tend to occupy regions of the slug associated with prestalk cell differentiation in chimera with wild-type. Schematic showing location of prestalk (pstA, pstO, pstB) and prespore (psp) cell types in the slug. Chimeric development of wild-type and *set1⁻* mutant cells. Genotypes of 90% unlabelled cells are shown in rows and genotypes of 10% actin-RFP expressing cells shown in columns. *Set1⁻* mutant cells sort to the collar and back of slugs when mixed with wild-type cells. Scale bar = 0.5 mm. (**B**) RNA-seq of FACS purified Set1

*Figure 6 continued on next page*

*Figure 6 continued*

mutant and wild-type cells. Actin15-GFP expressing wild-type cells were mixed with actin15-RFP expressing *Set1⁻* mutant cells in a 50:50 ratio. GFP and RFP cells were separated at the slug stage and subjected to RNA-seq. The cell type index (prestalk expression/(prestalk + prespore expression)) was calculated for genes differentially expressed in wild-type (blue bars) or Set1 mutant cells (yellow bars). >0.5 = prestalk,<0.5 = prespore. Interleaved histogram (light gray) shows the average distribution of 1000 random samples of the same numbers of genes. Scale bar = 0.5 mm. (**C**) Chimeric development of wild-type and set1 mutant cells expressing cell-type-specific markers. 10% labelled cells were mixed with 90% unlabelled cells. Set1 mutant cells strongly express pre-stalk markers, whereas a pre-spore marker is expressed weakly. Scale bar = 0.5 mm. (**D**) RNA-seq reveals Set1 mutant slugs exhibit defects in cell type proportioning. Clonal wild-type or *Set1* mutant cells were developed to the slug stage. The cell type index (>>0.5 = prestalk,<0.5 = prespore) was calculated for genes identified as differentially expressed by RNA-seq in Ax4 (blue bars) and *Set1-* (yellow bars) cells. Interleaved histograms (light grey) show the average distribution of 1000 random samples of the same numbers of genes. (**E**) Set1 mutant cells overexpress prestalk marker genes. Clonal wild-type or *Set1* mutant cells transformed expressing *ecmO*-lacZ- or *ecmB*-lacZ markers were developed to the slug stage. *ecmB* and *ecmO* are expressed in the collar and posterior anterior like cells in wild-type. In *Set1* mutant slugs, pre-stalk markers are expressed throughout the length of the slug. Scale bar = 0.5 mm. (**F**) Western blot of H3K4 methylation in Ash2 mutant cells. *ash2* mutant cells have lost H3K4 methylation. Levels of histone H3 are unaffected. (**G**) Western blot of H3K9/14 acetylation in Gcn5 mutant cells. *Gcn5* mutant cells have lost H3K9 and H3K14 acetylation. Levels of histone H3 are unaffected. (**H**) Chimeric development of Ash2 mutant cells with wild-type cells. Like set1⁻ cells, when labelled *ash2⁻* cells are mixed with WT cells they show a strong localisation in the collar and posterior of the slug. In the reciprocal mix, WT cells occupy the anterior pre-spore region. Scale bar = 0.5 mm. (**I**) Chimeric development of Gcn5 mutant cells. Like *Set1⁻* and *Ash2⁻* when labelled *Gcn5⁻* cells are mixed with WT cells they are localised to the collar and back of the slug. Scale bar = 0.5 mm.

The online version of this article includes the following source data and figure supplement(s) for figure 6:

**Source data 1.** PDF file containing original western blots for *Figure 6F*, indicating the relevant bands and treatments.

**Source data 2.** Original files for western blot analysis displayed in *Figure 6F*.

**Figure supplement 1.** Disruption of set1 does not affect developmental timing but results in slightly aberrant slug morphology.

**Figure supplement 2.** Set1⁻ mutant cells sort to upper and lower cup of fruiting bodies.

**Figure supplement 3.** A hypomorphic Set1 mutant exhibits a weaker developmental phenotype than knockout cells.

**Figure supplement 3—source data 1.** PDF file containing original western blots for *Figure 6—figure supplement 3A*, indicating the relevant bands and treatments.

**Figure supplement 3—source data 2.** Original files for western blot analysis displayed in *Figure 6—figure supplement 3A*.

being higher throughout the cell cycle in *set1⁻* mutant cells (*Figure 8C*). Finally, we tested whether the additive effects predicted by the stochastic-deterministic model are also seen during normal development. Developmental timing and fruiting body morphology is normal when cell cycle progression is perturbed in wild-type cells (by growth at 11.5 °C). Similarly, although changes in the level of expression of variable genes (through mutation of *set1*) affect proportioning, normal fruiting bodies are formed (*Figure 8D*). In contrast, when both inputs are simultaneously perturbed, the effects on development are dramatic. Few fruiting bodies formed, even 24 hr after development was complete in all other samples (*Figure 8D*). Together, these results are consistent with a model in which symmetry breaking, fate choice, and cell type proportioning in *D. discoideum* depends on the integration of two independent variable inputs. Since variation is widespread in biological systems, there is immense scope for natural selection to harness these properties for robust biological decision-making.

## Discussion

The ability of isogenic cells to break symmetry and adopt different fates allows unicellular organisms to cope with dynamically changing environments and underpins the division of labour, which is a key feature of the evolution of multicellularity (*West and Cooper, 2016*). Cell state heterogeneity is an inevitable feature of biological systems and can provide a reliable substrate for symmetry breaking. For example, stochastic intrinsic variation can be used to predictably 'sample' cells from different states if population sizes are sufficiently large (even if it is impossible to know what state each cell will be in). In other cases, cells may transition reproducibly between states, such as their position in the cell cycle. Moreover, if cell state reflects properties, such as energetic or resource state, it can be used to best determine the utility of cells for different roles.

### *Set1*-dependent methylation controls transcriptional burst frequency

Stochastic cell-cell variation can arise from noisy gene expression, which is an inevitable consequence of transcriptional bursting. Changes in burst frequency will affect the level of transcriptional cell-cell

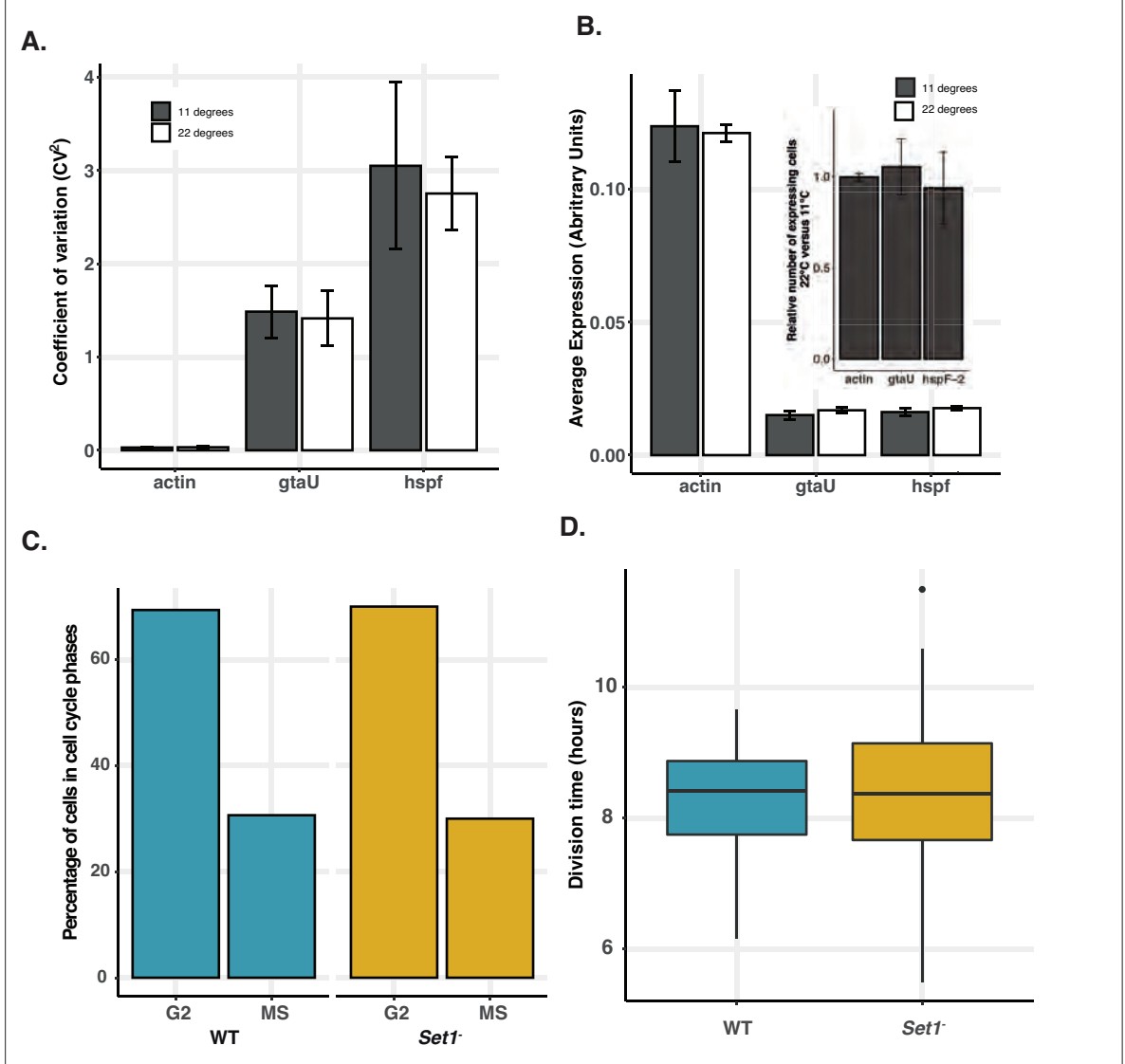

**Figure 7.** Cell cycle and stochastic variation are independent. (**A**) Variation in expression of Set1-dependent stochastic genes is unaffected by cell cycle disruption. Dual reporter lines for actin15, gtaU, and hspF-2 were incubated at either 22 °C or 11.5 °C and expression was quantified by FACS. (**B**) Level of expression of Set1-dependent stochastic genes is unaffected by cell cycle disruption. Dual reporter lines for actin15, gtaU, and hspF-2 were incubated at either 22 °C or 11.5 °C and average level of expression calculated. Inset shows comparison of proportion of positive fluorescent cells (above background) at 11 and 22°C. (**C**) Quantification of numbers of cells in different cell cycle phases. Single cells were assigned to different cell cycle stages based on the expression of MS and G2 markers from scRNA-seq (see *Figure 7—figure supplement 1* and methods). (**D**) Growth rate is unaffected by Set1 disruption. Cells were plated at low density and tracked by live imaging to determine cell cycle length of individual cells.

The online version of this article includes the following figure supplement(s) for figure 7:

**Figure supplement 1.** The cell cycle is unaffected in Set1 mutant cells.

variation, as well as mean expression levels. Bursting parameters and thus levels of transcriptional noise are affected by histone modifications. Roles in differentiation are suggested by studies in embryonic stem cells, as well as other systems suggest that gene networks associated with lineage choice are often associated with specific histone modifications (*Hong et al., 2011*; *Yadav et al., 2018*). This includes the presence of marks associated with gene activation (e.g. compass complex dependent H3K4 methylation and polycomb complex dependent H3K27 methylation). Single-cell RNA-seq analysis of mouse embryonic stem cells has also shown that treatment of cells with nucleoside analogues that are removed by the base excision repair pathway result in genome-gewide increases in cell-to-cell variability in transcript abundance of thousands of genes (*Desai et al., 2021*). Importantly, this

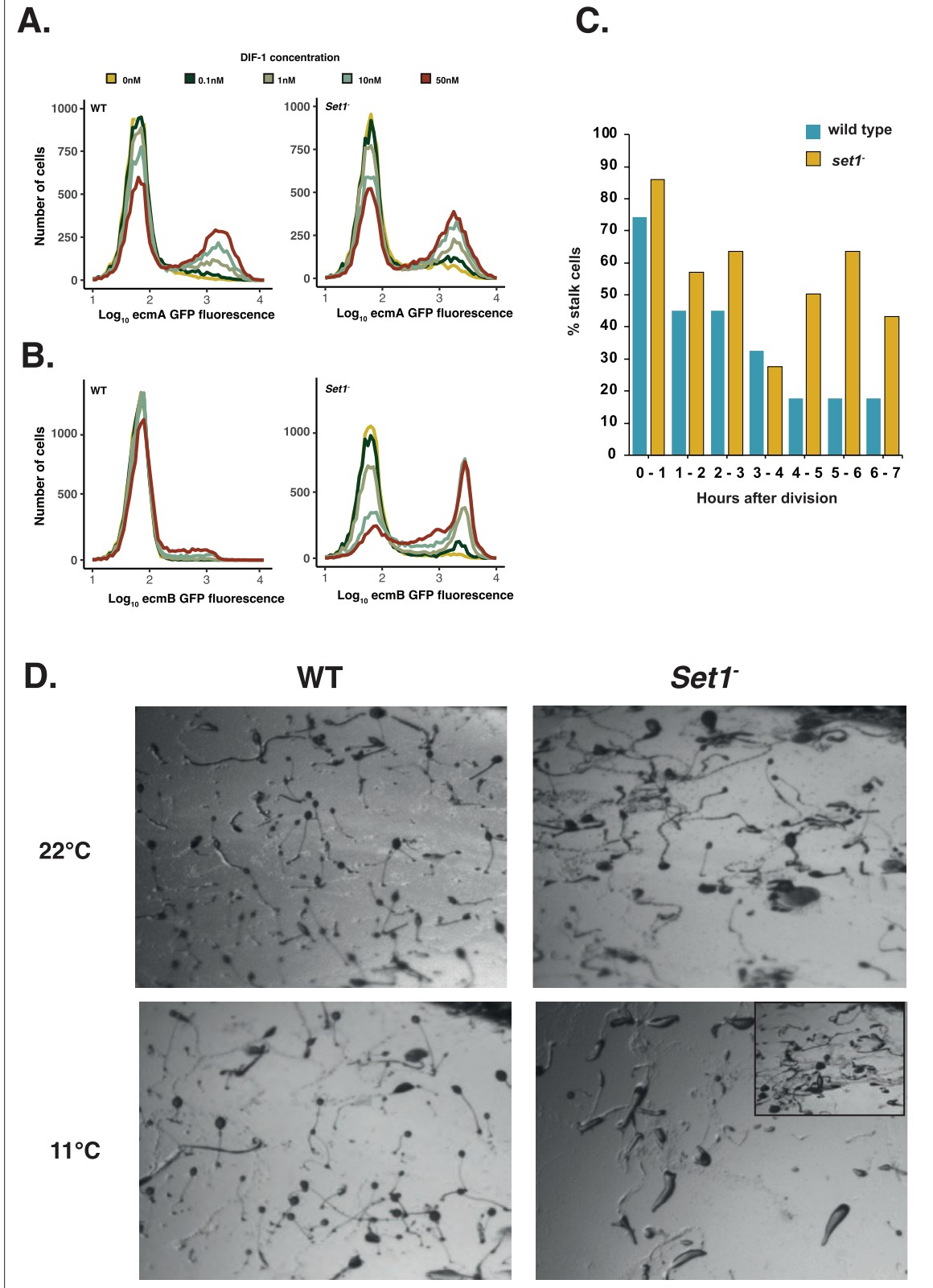

**Figure 8.** Cell type proportioning and responsiveness to differentiation cues depends on both stochastic and cell cycle variation. (**A** and **B**) Set 1 regulates the threshold of responsiveness to differentiation cues. *ecmA*-GFP (**A**) and e*cmB*-GFP (**B**) knock-in lines were generated in wild-type and *Set1*- mutant cells. Cells were induced to differentiate in culture in the presence of varying amounts of the prestalk inducer DIF-1. Marker gene expression was quantified by flow cytometry. Expressing cells were divided into bins of 0.05 (for log10 GFP expression). (**C**) Cell type differentiation at different cell cycle

*Figure 8 continued on next page*

*Figure 8 continued*

positions. Cells were grouped into hourly blocks according to the time of the last cell division before the addition of differentiation inducing signals. The percentage of cells that differentiate as prestalk cells was calculated. Results show the average of eight movies. (**D**) Development is blocked when stochastic and cell cycle gene expression is altered simultaneously. Wild-type (WT) and *Set1* mutant cells were incubated at 22 °C, or at 11.5 °C to inhibit cell cycle progression before being induced to develop. Scale bar = 1 mm.

The online version of this article includes the following figure supplement(s) for figure 8:

**Figure supplement 1.** Generation of ecmA and ecmB-GFP knock-in strains.

**Figure supplement 2.** Probability density functions for biased sampling from the cell cycle.

**Figure supplement 3.** Stochastic variation can buffer against changes in deterministic fate cues.

increase in transcriptional noise does not markedly change expression mean. This also affects the likelihood of differentiation, presumably because it changes the probability of cells being in a responsive state to fate-inducing signals. It is, however, unknown whether this reflects endogenous control mechanisms that regulate the levels of transcriptional noise.

Our results also support the idea that gene expression noise is important for cell fate. Our studies center on Set1-dependent H3K4 methylation, which is known to recruit various HATs, HDACs, and chromatin remodelers (e.g. ISWI, NURF, CHD) (*Bian et al., 2011*; *Santos-Rosa et al., 2003*; *Shi et al., 2006*; *Shi et al., 2007*; *Sims et al., 2005*; *Taverna et al., 2006*; *Wysocka et al., 2006*). Although Set1-dependent histone modifications are associated with more highly expressed loci (*Barski et al., 2007*; *Soares et al., 2017*), this relationship is not necessarily causal. In fact, genetic studies have revealed that absence of H3K4 methylation only affects a small subset of genes, and these are typically upregulated (*Hsu et al., 2012*; *Lorenz et al., 2014*; *Margaritis et al., 2012*; *Ramakrishnan et al., 2016*). In addition, previous studies have also noted a correlation between the breadth of the H3K4 methylation domain and the level of gene expression variability (*Benayoun et al., 2014*; *Rotem et al., 2015*; *Sze et al., 2020*; *Wu et al., 2017*). Indeed, disturbance of these patterns affects transcriptional consistency (*Sze et al., 2020*; *Wu et al., 2017*). Our studies provide further support for these ideas. Moreover, they support the idea that the modification of chromatin structure provides a means by which the level of gene expression and gene expression variation can be controlled to facilitate cell fate decisions.

## Noisy gene expression has the potential to buffer development against environmental perturbations

Cell fate choice and symmetry breaking in *D. discoideum* depend on the interplay between deterministic effects of cell cycle position and stochastically expressed developmental genes on responsiveness to fate-inducing signals. Why should such a mechanism evolve when a purely stochastic system could achieve correct proportioning simply by random sampling of a sufficiently large population of cells? Similarly, a cell cycle system alone could also achieve correct proportioning if cells are randomly distributed through the cell cycle. *D. discoideum* cells that have recently completed mitosis (i.e. those that have recently divided their resources) are biased towards the stalk fate. Use of the cell cycle position presumably reflects an adaptation that helps ensure aggregations to disproportionately sacrifice 'lower value' cells to the stalk for building the stalk and benefit higher value (higher resource state) cells that form spores. Resource-based division of labour is also seen in *Sinorhizobium meliloti*, where cells with highest levels of poly-3-hydroxybutyrate (PHB) (which allows survival in long-term starvation) do not grow and instead leave resources to be utilised by low PHB-containing cells (*Ratcliff and Denison, 2010*).

While such examples suggest cell state heterogeneity can be utilised adaptively to make cellular decisions, it is also vulnerable to systematic perturbations that affect the proportion of cells in different states. As soil dwellers, *D. discoideum* cell populations are likely to be exposed to varying environmental conditions, such as temperature, pH, or available nutrients, that can all potentially affect developmental signalling, cell physiology, and cell type proportions. For example, *D. discoideum* cells depleted in glucose or that have experienced cold shock will arrest around mitosis (and end up biased towards becoming stalk), which means that certain environments could lead to extreme stalk fate bias. Therefore, the evolutionary success of this organism means that mechanisms likely evolved to ensure development is buffered against environmental perturbations. Early studies identified a mechanism

by which cell type proportioning could be corrected by feedback loops where the excess of one cell type leads to its trans-differentiation into the other (*Belcher et al., 2022*; *Insall et al., 1992*; *Kay and Thompson, 2001*). It has also been shown that disturbances that affect the cell cycle result in compensatory gene expression changes that alter the threshold of responsiveness (*Chattwood et al., 2013*). Our findings provide another potential solution to this problem, in which the integration of different sources of cell state variation allows systems to be robust against environmental perturbations, while still being able to adaptively exploit differences in cell state. This is because stochastic cell-cell variation could buffer against catastrophic shifts in cell-type proportioning, by ensuring some cells always adopt different fates (*Figure 8—figure supplements 2 and 3*). This would reduce the range over which feedback loops that refine proportioning need to operate. It is tempting to speculate that this complexity is a consequence of the strength of selection to ensure the correct ratio of stalk and spore cells under fluctuating natural environments that may affect signalling and cell fate choice (e.g. temperature, pH, light, salt levels, humidity, food availability). Moreover, it highlights the impacts that cell-cell variation and environmental variation may have on the evolution of developmental signalling pathways. Since variation can promote or hinder developmental robustness, it is likely that many developmental systems will exhibit the Goldilocks principle when it comes to adaptive exploitation of heterogeneity, where they have been evolutionarily tuned so that they are reliably and repeatedly exposed to 'just the right amount' of variation.

## Methods

### Strains, culture, and development

*Dictyostelium discoideum* strains were derived from parental *Ax3* or *Ax4* strains. Cells were cultured in 10 cm tissue culture dishes containing HL5 media including glucose (Formedium) or on SM agar plates (Formedium) supplemented with *Klebsiella aerogenes*. For development, cells in tissue culture were diluted to $1\times10^5$ cells/ml and allowed to grow for 2–3 divisions before harvesting. Cells were induced to develop at a density of $5.1\times10^5$ cells/cm$^2$ on non-nutrient KK2 agar (16.1 mM $KH_2PO_4$, 3.7 mM $K_2HPO_4$) plates containing 1.5% purified agar (Oxoid). For chimeric development, cells were mixed prior to spreading on agar. *Set1* mutant lines were generated using a knock-out construct kindly provided by J. Chubb. *Ash2 and Gcn5* mutants were selected from the GWDI project (https://remi-seq.org) and verified by diagnostic PCR. A strain with a random intergenic insertion (GWDI_42_D_7) was utilised as a control. For cell cycle synchronisation by cold shock, exponentially growing cells were seeded at $1.5\times10^6$ cells/ml and incubated for 20 hr.

### Reporter gene analyses

To measure cell-cell variation in gene expression of stochastic genes, promoter sequences 1 kb upstream of the start codon were amplified and cloned into pDM324 (*Veltman et al., 2009*). The vector pDM327 was digested with NgoMIV to add sequences containing the *actin15*-promoter-GFP-*actin8*-terminator. *ecmA*-GFP or *ecmB*-KI GFP knock-in strains were generated by gene replacement (*Chattwood et al., 2013*). To quantify prestalk cell differentiation by FACS, growing cells were re-suspended at $1\times10^5$ cells/ml in 1.5 ml of 1 x stalk solution supplemented with 5 mM cAMP and incubated in 35 mm plastic bottom dishes for 9 hrs at 22 °C. Inductions were carried out for a further 9 hr at 22 °C by replenishing the cAMP and adding different doses of DIF-1. For analysis, cells were washed and collected in 1 ml of KK2+20 mM EDTA. Fluorescence was measured by FACS (Nxt Flow Cytometer (Attune)). Relative proportion of RFP and GFP-positive was assessed using FlowJo software. LacZ staining of developed structures was carried out using strains expressing β-galactosidase under promoter control of fate-specific markers ecmO and ecmB. Strains were fixed for 10 min in 4% formaldehyde and Z-buffer (10 mM KCl, 60 mM Na2HPO4, 40 mM NaH2PO4, 1 mM MgSO4). Structures were washed twice in Z-buffer before being permeabilised for 20 min in 0.1% NP40. Another two washes were carried out before adding staining solution (5 mM K3Fe(CN)6, 5 mM K4Fe(CN)6, 1 mg/ml X-gal, 1 mM EGTA).

### Western blotting

Growing cells were lysed in nuclei buffer (40 mM Tris-pH7.8, 1.5% Sucrose, 0.1 mM EDTA, 6 mM $MgCl_2$, 40 mM KCl, 5 mM DTT, 0.4% NP40). Pellets were re-suspended at a density of $1\times10^7$ nuclei/

ml in 1 x PBS, including 1 x protease inhibitors (Promega #G6521) and 1 X SDS-PAGE buffer. Western blots were probed with antibodies to Histone-3 (ab1791 - polyclonal-rabbit IgG), H3K4-me1 (ab8895 - polyclonal-rabbit IgG), -me2 (Millipore #07–030 - polyclonal-rabbit IgG), -me3 (ab8580 - polyclonal-rabbit IgG), β-actin (Santa-Cruz Biotechnology – monoclonal-mouse IgG). Gcn5[-] blots were carried out as previously described (*Huang et al., 2021*). Blots were imaged using a Chemidoc MP imaging system (Biorad) and Image Lab 5.2.1 software (Biorad).

## Single cell analysis of cell cycle and cell fate

To follow differentiating cells, cells were transformed with the *pspA*-GFP cell-type-specific reporter gene. Log phase cells were resuspended in HL5 glucose media and $7.5 \times 10^3$ cells were plated in 750 µl deposited on a 35 mm glass bottom dish (WPi). Cells were grown for a minimum of 16 hr during which time cells were imaged every 5 min. The time between cell divisions was determined by manually tracking single cells. After 16 hr, HL5 media was removed and cells were washed five times in KK2 buffer to remove residual growth medium. To induce differentiation, 750 µl of conditioned media supplemented with 10 nM DIF-1 was added. Conditioned media was collected from cells developed at a high density ($1 \times 10^6$ cells/ml) in 1 x stalk solution (10 mM MES pH 6.2, 1 mM CaCl$_2$, 2 mM NaCl, 10 mM KCl, 200 mg/ml streptomycin sulphate) supplemented with 5 mM cAMP for 16 hr. Conditioned media was collected and briefly centrifuged before DIF-1 was added to washed cells. Images were taken at 5 min intervals for another 20 hr. A final fluorescence image was taken to determine the number of prestalk and prespore cells. Cells were tracked by hand to determine the timing of the last cell division prior to the induction of cell differentiation.

## REMI-seq screen for DIF-insensitive mutants

A REMI mutant library consisting of >10,000 mutants (*Gruenheit et al., 2021*) was grown to log phase. Cells were plated in triplicate at $2 \times 10^5$ cells/ml in stalk medium in 10 cm diameter tissue culture dishes with 5 mM cAMP for 24 hr. Cells were then washed twice with KK2 before 24 hr incubation with 10 nM DIF. Stalk medium was then removed and replaced with growth medium (HL5) and cells were allowed to grow until reaching confluency. Genomic DNA was prepared from the mutant library following 2 and 6 rounds of this selection and processed for sequencing. To control for mutants that are unable to differentiate in response to cAMP, an 8-Br-cAMP monolayer assay screen was performed. Cells from the mutant library were seeded in triplicate at $2 \times 10^5$ cells/ml in stalk medium supplemented with 10 mM 8-Br-cAMP. After 48 hr, detergent was added (0.1% NP40, 10 mM EDTA) to remove cells that had not formed spores. Stalk medium was then removed and replaced with growth medium (HL5) and the cells were grown until reaching confluency. Genomic DNA was prepared from the mutant library following 1 round of this selection and processed for sequencing (*Gruenheit et al., 2021*). Analysis of mutant pools was carried out as previously reported (*Gruenheit et al., 2021*) using Z-score thresholds of >1.5 for enriched mutants and <–1 for depleted mutants. Mutants (see Supplementary data) were identified as DIF-insensitive if they were enriched in the cAMP removal screen, not down-regulated in 8-Br-cAMP screen and not previously identified as having a growth phenotype (*Gruenheit et al., 2021*).

## Sample preparation and analysis of bulk RNA-seq data

RNA was extracted from log-phase growing cells or cells developed for 16.5 hr on 1.5% non-nutrient agar at 22 °C. For RNA sequencing of chimeric development, wild-type cells expressing *actin15*-GFP were mixed in a 1:1 ratio with *set1*[-] cells expressing *actin15*-RFP and developed for 16.5 hr. Chimeric slugs were disaggregated by passing through a 25 G needle before sorting (BD FACSaria) into GFP and RFP positive populations. RNA was extracted and RNA integrity number (RIN) of samples was determined by Tapestation (Agilent). Libraries were prepared from samples deemed with a RIN >8 using an Illumina TruSeq kit and sequencing was undertaken on a HiSeq-4000 (Illumina) using 100 bp pair-end chemistry. Sequences were trimmed of TruSeq adapters and quality controlled (Trimmomatic) by discarding reads shorter than 20 bp or those where the average quality score dropped below an average of 15 in a sliding four base pair window. Leading and trailing bases of reads below a phred score of 30 were also removed from tags. Reads were aligned (Bowtie2) to the *D. discoideum* genome an inverted repeat on chromosome 2 was masked, bamfiles were sorted (Samtools) and reads counted using the RPKM_count.py script (RSeQC). DESeq2 v1.26.0 was used for differential expression

analyses. Thresholds for differential expression between samples were set at a *p.adj*<0.01 with a fold change of >2 between samples. To calculate differences in cell-type-specific gene expression, prestalk, and prespore RNA-seq data was downloaded from the SRA (PRJNA543665) (*de Oliveira et al., 2019*). Genes with less than 10 read counts were removed. A cell type index was calculated for the remaining 5319 genes (Cell type index = expression count in prestalk cells/expression count in prestalk cells + expression count in prespore cells).

## ChIP-seq analysis

Bulk RNA-seq data from vegetative cells (this study) was used to rank genes based on their level of expression. Genes with detectable expression (i.e. >0 normalised reads) were divided into ten equally sized bins. ChIP-seq data for two H3K4me1 and H3K4me3 replicates was downloaded from the GEO database (accession #GSE137604 Sub-Series GSE137599) as narrowPeak files (*Wang et al., 2021*). Promoter regions around the TSS for each gene in each bin were identified (–2500 bp to +2500 bp up/down-stream of the TSS) and annotated. Using functions from the chIPSeeker package and custom scripts (https://github.com/WilliamSalvidge/dictyChipSeq, *Salvidge, 2023*) annotated regions were intersected H3K4-me1 or -me3 peaks as defined by narrowPeak files. Overlaps were averaged for each expression bin and plotted using the *plotAvgProf* function from the chIPSeeker R package. This accounts for differing numbers of peaks in different expression bins and allows patterns of peak density between expression bins to be compared. To compare peak distribution in variable and non-variable genes, an equal number (2024) of genes was sampled using a weighted probability based on the expression of variable genes.

## Identification of variable genes using single-cell RNA-sequencing

Data for 81 single wild-type cells isolated using the Fluidigm C1 platform were downloaded from the SRA (SAMN07833758 - SAMN0783383). Reads were normalised (DESeq2 v1.26.0) and the coefficient of variation squared (CV$^2$) was calculated and plotted against mean expression. A trend line was fitted to the data using non-linear least squares regression (Scran v1.15.9). Genes were defined as variable (2073 genes) based on a one-sided test assuming a normal distribution around the trend but one where deviation changed depending on the mean expression of a given gene (Scran v1.15.9 - model-GeneCV2) with a FDR of <0.05. To identify genes with a cell cycle signature, single cells were clustered using M3Drop v1.12.0 as previously reported (*Gruenheit et al., 2018*). Cells could be organised into three clusters of 25, 31, and 25 cells. Marker genes for each cluster were identified using logistic regression (M3Drop v1.12.0). A relaxed AUROC threshold (>0.65) ensured that all genes possessing weak cell cycle signature could be identified (5529 genes). This allowed variable genes to be identified that where variation is dependent on cell cycle position (1016 genes) and independent of cell cycle position (1057 genes).

## Single-cell sequencing of wild-type and *set1*⁻ cells

*Actin15-GFP* expressing wild-type or *set1*⁻ cells were grown on tissue cultures dishes and harvested during log-growth. Cells were washed in 1 x PBS and incubated with DAPI (2.5 µg/ml). Cells were resuspended at a density of 2.8×10⁴ cells/ml and dispensed into a SMARTer ICELL8 3' DE Chip using the ICELL8 cx Single Cell System. Wells of the 3' DE Chip contain pre-printed oligonucleotides possessing well-specific barcodes and UMI connected to a polydT region for hybridisation with polyadenylated transcripts. For each well of the chip, 50 nl of stained cell solution was aliquoted to maximise the number of wells containing a single cell. Cells were imaged directly dispensing into nano-wells using DAPI and FITC filters and the 3' DE Chip was frozen at –80 °C. Images taken were analysed to identify wells containing individual cells based on DAPI and GFP fluorescence. In total, 799 wells were identified that contained single cells (399 wild-type and 400 *set1*⁻). The 3' DE Chip was then thawed to lyse cells and loaded onto the ICELL8 cx Single Cell System, where components for reverse transcription (RT) and cDNA amplification were dispensed into chosen wells. After RT-PCR, products from separate wells were collected into a single sample, concentrated and purified according to the manufacturer's instructions. Samples were prepared for sequencing using a Nextera XT Library Prep Kit (Illumina) and sequenced on a NextSeq 500 (Illumina) system utilising one flow cell and a NextSeq High-Output kit (2×75 bp reads). One read was used to sequence the well barcode and transcript UMI, with the second reading the 3' end of the transcript itself. This yielded >500 million reads. FASTQ files were

demultiplexed (mappa), reads were then quality controlled (reads shorter than 15 bp discarded, a minimum of 30% N's allowed, phred score of 20) and trimmed of adapters (cutadapt). Reads were aligned (STAR) to the latest version of the *D. discoideum* genome (v2.7), sorted (Samtools) and tags counted (UMI-tools). Cells were quality controlled (Scater v1.14.6) and cells over 2 median associated deviations (MADs) from the median for library size, total number of features or mitochondrial reads excluded as outliers. This left 310 wild-type and 310 *set1⁻* cells. Genes with <5 reads were also removed from further analyses.

## Cell cycle analysis

Genes previously defined by scRNA-seq as markers of M/S and G2 cells from 81 Fluidigm-sorted cells (*Gruenheit et al., 2018*) were used to compare cell cycle patterns in iCell8-sorted Ax4 and *Set1⁻* cells. The average expression of all 876 M/S genes and 642 G2 genes with expression in these samples (>0 reads) was determined. The normalised ratio of M/S to G2 marker gene expression was used to define the cell cycle position in each cell.

## Model fitting to measurements of stalk cell induction

To evaluate the fit of our stochastic-deterministic model to data, we used data on stalk fate measured at different times in the progression through the cell cycle in population of cells aligned in the cell cycle (*Gruenheit et al., 2018*). Stalk fate propensity was measured in two genetically different sets of cells (wildtype AX3 and AX3 with a knockout of the gene *gefE*) grown under two conditions ('normal', G+, and low glucose, G−, conditions), which alter the stalk propensity of cells. We used data from the first six timepoints, corresponding to 0–5 hr after the last division, and combined the four sets of cells by adjusting the values in set such that their mean propensity matched the overall global mean propensity (and hence there is no difference in average propensity of the four sets; see Supplementary Data for the raw and adjusted values). After combining the four sets, one outlier was identified (*gefE⁻* under G− at 5 hr), which was consistent with those cells passing a checkpoint where they re-enter mitosis, and was removed, after which data were re-normalised as described above (see Appendix 1 for a comparison of the model fitting with the outlier included). To confirm that the different sets of cells behave similarly, we also fitted the model separately to each class and see no evidence of heterogeneity of model estimates. The stochastic-deterministic model was fitted to these data using the 'NonlinearModelFit' function in Wolfram Mathematica version 14, which uses the Levenberg–Marquardt algorithm for least-squares curve fitting. This model fitting yielded estimates of $C_0^*$ and $\beta$, the Akaike Information Criterion (AIC), and of the error and total sums of squares, which were used to calculate the *R*-squared. The details of the alternative models that were fitted and the methods used for comparing the fit of different models to the stalk propensity data are provided in Appendix 1.

## STAR methods
### Software

**Trimmomatic** - http://www.usadellab.org/cms/index.php?page=trimmomatic
**Bowtie2** - http://bowtie-bio.sourceforge.net/index.shtml
**Samtools** - http://samtools.sourceforge.net
**RSeQC -** http://rseqc.sourceforge.net
**Takara mappa** - https://takarabiousa.github.io/mappa_userguide.html
**Cutadapt -** https://github.com/marcelm/cutadapt (*Martin et al., 2026*)
**STAR aligner manual** - https://github.com/alexdobin/STAR/blob/master/doc/STARmanual.pdf (*Dobin, 2024*)
**UMI tools manual** - https://github.com/CGATOxford/UMI-tools (*CGATOxford, 2026*)
***D. discoideum genome -*** https://protists.ensembl.org/Dictyostelium_discoideum/Info/Index
**Mathematica 14.0** - https://www.wolfram.com/mathematica/ (*Wolfram, 2024*))

### R - packages

**DESeq2 - v1.26.0** - https://bioconductor.org/packages/release/bioc/html/DESeq2.html
**Scater - v1.14.6** - https://bioconductor.org/packages/release/bioc/html/scater.html

**Scran - v1.15.9** - https://bioconductor.org/packages/release/bioc/html/scran.html
**M3Drop - v1.12.0** - https://bioconductor.org/packages/release/bioc/html/M3Drop.html
**SC3 - v1.14.0** - https://www.bioconductor.org/packages/release/bioc/html/SC3.html
**Superheat - v.0.1.0** - https://rlbarter.github.io/superheat/ (*Barter, 2017*)

## Acknowledgements

This work was supported by a Wellcome Trust Investigator Award (WT095643AIA) to CRLT and grant from NERC (NE/V012002/1) to CRLT and JBW.

## Additional information

### Funding

| Funder | Grant reference number | Author |
| --- | --- | --- |
| Wellcome | 10.35802/095643 | Chris Thompson |
| Natural Environment Research Council | NE/V012002/1 | Jason Wolf Chris Thompson |

The funders had no role in study design, data collection and interpretation, or the decision to submit the work for publication. For the purpose of Open Access, the authors have applied a CC BY public copyright license to any Author Accepted Manuscript version arising from this submission.

### Author contributions

William Salvidge, Conceptualization, Data curation, Formal analysis, Supervision, Investigation, Methodology, Writing – original draft, Writing – review and editing; Chris Brimson, Supervision, Investigation, Methodology, Writing – review and editing; Nicole Gruenheit, Data curation, Formal analysis, Investigation, Methodology; Li-Yao Huang, Data curation, Investigation; Catherine Pears, Investigation; Jason Wolf, Conceptualization, Formal analysis, Investigation, Methodology, Writing – original draft, Writing – review and editing; Chris Thompson, Conceptualization, Formal analysis, Supervision, Funding acquisition, Investigation, Visualization, Methodology, Writing – original draft, Project administration, Writing – review and editing

### Author ORCIDs

Jason Wolf ⬥ https://orcid.org/0000-0003-3112-6602
Chris Thompson ⬥ https://orcid.org/0000-0002-5240-2592

Reviewer #1 (Public review): https://doi.org/10.7554/eLife.105512.3.sa1
Reviewer #2 (Public review): https://doi.org/10.7554/eLife.105512.3.sa2
Author response https://doi.org/10.7554/eLife.105512.3.sa3

## Additional files

### Supplementary files
MDAR checklist

Supplementary file 1. Supplementary data.

### Data availability
RNA sequence data were deposited in GEO under project codes PRJNA1034001 and PRJNA1037558. All data generated or analysed during this study are included in the manuscript and supporting files; source data files have been provided for all figures.

The following datasets were generated:

| Author(s) | Year | Dataset title | Dataset URL | Database and Identifier |
|---|---|---|---|---|
| Salvidge WF, Brimson C, Gruenheit N, Huang L-Y, Pears CJ, Wolf JB, Thompson CRL | 2026 | Harnessing noise to enhance robustness of deterministic developmental signalling | https://www.ncbi.nlm.nih.gov/bioproject/PRJNA1034001 | NCBI BioProject, PRJNA1034001 |
| Salvidge WF, Brimson C, Gruenheit N, Huang L-Y, Pears CJ, Wolf JB, Thompson CRL | 2026 | Single cell sequencing of Dictyostelium discoideum AX4 and Set1 knock-out cells | https://www.ncbi.nlm.nih.gov/bioproject/PRJNA1037558 | NCBI BioProject, PRJNA1037558 |

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

## Appendix 1

### Expectation of the stochastic-deterministic model at its limits

In the main text, we note that the stochastic-deterministic model can produce anywhere from the step-like function expected under the deterministic model (*Equation 1* and *Figure 1A*) and the constant proportioning expected for a purely stochastic model (*Equation 2* and *Figure 1B*). The case of the purely deterministic model can be derived by taking the limit where variation from noisy gene expression vanishes, which occurs as the variance in gene expression goes to zero (i.e. as σ→0):

$$P_{t(\text{deterministic})} = \frac{1}{2}\left(1 + \frac{\sqrt{(C_0 - R - \beta t)^2}}{C_0 - R - \beta t}\right) \qquad (S1)$$

This expression takes on the value of 1 during the period of the cell cycle where $t < (C_0 - R)/\beta$ and a value of 0 otherwise. The case of the purely stochastic model can be derived by taking the limit where the cell-cycle dependent component vanishes (i.e. as $\beta \to 0$), which yields the expression given by *Equation 2*, where stalk propensity depends on the properties of the stochastically expressed genes ($\mu$ and $s$) relative to the threshold that results in stalk fate ($R$).

### Comparison of model fit to the exponential-decay model of Gruenheit et al.

#### Model fitting methods

In the main text we evaluate the fit of the stochastic-deterministic model to the stalk propensity data from *Gruenheit et al., 2018*, and compare it to their exponential-decay model. Here, we provide further information on the exponential-decay model and the methods used to compare the fit of the two models.

Like the stochastic-deterministic model, the exponential-decay model is a two-parameter model that predicts stalk propensity through the cell cycle. It treats stalk propensity as a deterministic property that changes through the cell cycle according to the equation $\tilde{P}_t = \chi \exp(-\lambda t)$, where $\chi$ represents the starting stalk propensity (just after the end of mitosis), $\lambda$ the exponential rate of decay in stalk propensity, and, $t$ the time since the end of the last mitosis (with the tilde being used to differentiate the expectation for this model from the expectation from the stochastic-deterministic model). Although both models have two parameters that capture similar properties, they are not equivalent to each other. The stochastic-deterministic model defines a starting stalk propensity as a Z-score ($C_0^*$) that translates into the proportion of cells from a stochastically variable (Gaussian) distribution that will become stalk, whereas the exponential-decay model defines a fixed starting propensity ($\chi$). Both models have decay terms, with the decay term in the exponential-decay model ($\lambda$) giving the exponential rate of decrease in stalk propensity, while the decay term in the stochastic-deterministic model ($\beta$) represents the decay in the standardised mean CCAF level.

The exponential-decay model was fitted to the data following the same method as that described in the text for the fitting of the stochastic-deterministic model. Briefly, the model was fitted using the 'NonlinearModelFit' function in Wolfram Mathematica version 14, which uses the Levenberg–Marquardt algorithm for least-squares curve fitting. This generated estimates for the two parameters, $\chi$ and $\lambda$ and for the Akaike Information Criterion (AIC) of the fitted model. We used the error and corrected total sums of squares to calculate the $R$-squared (since it is calculated incorrectly by default in the NonlinearModelFit function).

We evaluated the relative support for each model by comparing the Akaike weights (*Burnham et al., 2002*; *Wagenmakers and Farrell, 2004*), which reflect the relative likelihood of the models. To further evaluate the relative fit of the models, we used the parameter estimates to generate a set of predicted values for each model and treated these as independent variables in a single linear model where the observed (adjusted) stalk propensity values were the dependent variable, which essentially allows the two models to directly compete for the fit to the data.

## Model fitting results

As expected based on the findings of *Gruenheit et al., 2018*, the exponential-decay model provides a good fit to the data (adjusted *R*-squared=0.88, *AIC* = −47.19), yielding an estimated rate of decay of $\lambda = 0.351$ and a starting stalk propensity of $\chi = 0.74$ (*Figure 1—figure supplement 1Ai*), which would correspond to an overall steady-state stalk propensity of 0.35 (where all times since the end of the previous mitosis are sampled equally). As noted in the main text, although the model provides a good fit to the data, it does not offer a mechanism to translate CCAF levels into quantitative variation in stalk propensity. Hence, it provides a description of the pattern of change in stalk propensity through the cell cycle (as being approximately an exponential decline) rather than a mechanistic model for the process that produces the predicted value of stalk propensity.

The ratio of the Akaike weights of the stochastic-deterministic model compared to the exponential-decay model has the value 98.35, which implies that the stochastic-deterministic model is 98.35 times more likely (in terms of Kullback–Leibler discrepancy). This result can also be interpreted as a probability of 0.990 that the stochastic-deterministic model is the better fitting model (meaning we can reject the exponential-decay model as the better fitting model with *p*=0.010). This conclusion is further supported by the results of the linear model that competed the two sets of predictions, which indicated that the partial fit to the predicted values from the stochastic-deterministic model is very good ($F_{1,20}$ = 9.83, *p*=0.0052) while the fit to the predicted values from the exponential-decay model is not significant ($F_{1,20}$ = 0.48, *p*=0.49).

## Evaluating the assumption of normally distributed noise

To evaluate the assumption of normality for the distribution of noise in the stochastic-deterministic model, we used an approach that models noise based on the gamma distribution, which is a family of continuous distributions that vary in the values of two parameters (in our implementation, these are a shape parameter *k* and a scale parameter $\theta$). By varying the two parameters, the gamma distribution provides probability density functions that go from a pattern of exponential decay to a pattern that approximates normality. The gamma distribution does not have a fixed mean and variance, but the shape parameters together dictate these two properties of the distribution, where the mean is given by $k\theta$ and variance by $k\theta^2$. Using these properties, we modelled decay in CCAF by adapting the gamma distribution to a form that is directly analogous to the approach implemented in the stochastic-deterministic model:

$$P_t = \frac{\theta^{-k}}{\Gamma(x)} \int_R^\infty e^{-\frac{x}{\theta}} x^{k-1} dx$$

$$= \frac{\Gamma\left(k, \frac{C_0 + \beta t}{\theta}\right)}{\Gamma(k)} \tag{S2}$$

where $\Gamma(x)$ is the Euler gamma function, and $\Gamma(x, y)$ is the incomplete gamma function (which is integrated from *y* to ∞).

Following the methods used to fit the exponential-decay and stochastic-deterministic models, we again used the 'NonlinearModelFit' function in Wolfram Mathematica version 14 to fit the gamma-distribution model (*Equation S2*) to our data, except the model includes a total of four parameters ($C_0$, $\beta$, *k*, and $\theta$) instead of two parameters like the other two models. This not only allows us to evaluate whether the best fit function was consistent with the assumption of normality, but also to further evaluate the Gaussian noise model against the exponential-decay model (since the best fit gamma distribution could approximate either distribution). We find that the best fit model ($C_0$=1.227, $\beta = 0.074$, $k = 56.9$, and $\theta = 0.023$) has a good fit to the data (adjusted *R*-squared=0.91, *AIC* = −51.60; see *Figure 1—figure supplement 1Aii*), with a gamma distribution component corresponding to a distribution of noise that is approximately normal (*Figure 1—figure supplement 1Bi and ii.*). Hence, the additional two parameters for the gamma distribution yield a similar fit to the simpler stochastic-deterministic model that assumes Gaussian noise, meaning that gamma-distribution model will necessarily be the poorer fitting model given it has two additional parameters (so we do not include any formal model comparisons). This conclusion is further supported by the ratio of the Akaike weights of the model with Gaussian noise compared to the model with gamma

distributed noise, which has the value 10.85, meaning that, despite having two fewer parameters, the Gaussian noise model is 10.85 times more likely (corresponding to a probability of 0.92).

## Evaluating the assumption of linear decay in CCAF

The simple version of the stochastic-deterministic model presented in the main text assumes that CCAF shows a linear decay through the cell cycle. What this means in terms of the model properties is that the mean of the distribution of stalk-inducing factors changes linearly through the cell cycle (since the level of CCAF determines the mean level of stalk-inducing factors, while CCIF adds variability around this mean). Importantly, this assumption does not necessarily imply that there is a linear change in some factor(s) at the molecular level (since the number of molecules of a signal could decay exponentially, but the effect of the signal could decay linearly if there is a non-linear relationship between signal levels and their effect, and vice versa). For many biological processes, we might expect factors to show non-linear decay, especially exponential decay (e.g. if concentrations showed a constant half-life). Therefore, to evaluate whether a model with non-linear decay provides a better fit to the data, we replaced the linear decay function in *Equation 1* with an exponential decay function. Initial attempts to fit a simple two-parameter decay process (such as the exponential decay equation fitted in Gruenheit et al., which we could write as $A_0 e^{-\delta t}$) made it clear that there was no parameter space that would produce the pattern of change in stalk proportioning observed through the cell cycle. This issue is due to the fact that the simple exponential decay equation ties together the 'starting value' (which would be $A_0$ when $t = 0$) and the effective rate of decay (e.g. the derivative of the decay function with respect to time, would be $-\delta A_0 e^{-\delta t}$, so the linear component depends on the decay rate $\delta$ scaled to the starting value $A_0$). Therefore, we used a more flexible exponential decay function that allows for separation between the starting value and the rate of decay:

$$A_t = A_0 - A^* e^{-\delta t} \tag{S3}$$

The structure of *Equation S3* is superficially similar to the linear decay equation (*Equation 2*). However, while $A_0$ is the starting level of CCAF in *Equation 1* (i.e. the level of CCAF that determines stalk proportioning at the start of the cell cycle), in *Equation S3*, the starting level (where $t = 0$) would be $A_0 - A^*$. The level of CCAF is eroded by an exponential process captured by the second term on the RHS, which has two parameters, $A^*$, which represents the size of the CCAF pool that decays, and $\delta$, which gives the rate of exponential decay.

Following the same logic outlined in the main text for the case of linear decay in CCAF, we assume stalk proportion depends on the proportion of cells experiencing a value of CCAF+CCIF that is above a threshold value. Replacing linear decay with the exponential decay process in *Equation S3* gives $C_t = \mu + A_0 - A^* e^{-\delta t} = C_0 - A^* e^{-\delta t}$. As in the linear decay model, we can rescale the value of $C_0$ relative to $R$, though the interpretation of the resulting parameter $C_0^*$ is different in this case since it does not, on its own, capture the baseline value that represents the expectation for stalk propensity at the start of the cell cycle. Instead, while $C_0^*$ gives the baseline value in the linear decay model (where $C_0^* = R - (\mu + A_0) = R - C_0$), in the exponential decay model, the baseline value (i.e. the value that determines stalk fate at the start of the cell cycle) is given by $C_0^* - A^*$. Using this value of $C_0^*$ in place of the one for the linear model to derive the stalk propensity gives (*Equation 3*):

$$P_t = \frac{1}{2}\left(1 - e^{\delta t}\sqrt{e^{-2\delta t}}\right) erf\left[\frac{\sqrt{e^{-2\delta t}}\left(A^* - C_0^* e^{\delta t}\right)}{\sqrt{2}}\right] \tag{S4}$$

Because of the more complex structure of *Equation S4*, this model was fitted using the 'NMinimize' method in the 'NonlinearModelFit' function in Wolfram Mathematica version 14, which forces a global search for the best fit parameters and helps avoid local minima (note that using this method for all other models fitted in our study has no impact on the estimates). The exponential decay model shows a slightly worse fit than the linear decay model (adjusted $R$-squared=0.920 and AIC = −56.37 for the linear decay model, while adjusted $R$-squared=0.916 and AIC = −54.24 for the exponential decay model), but given that the fits are almost identical, and the linear decay model shows a slightly better fit, the linear decay model is necessarily the more likely model (with the ratio of the Akaike

weights being 2.90). Moreover, the two models produce very similar estimates for the starting level of CCAF (which is an estimate of $C_0^*$ in the linear decay model and $C_0^* - A^*$ in the exponential decay model), with values of 0.57 for the linear model and 0.56 in the exponential model. They also produce similar estimates for the linear rate of decay (which is given by the term $\beta$ in the linear model, and the first derivative of the solution with respect to $t$ in the exponential decay model), with values of 0.41 for the linear model and 0.39 for the exponential model. Importantly, the estimates from the exponential decay model produce a pattern of decay in CCAF that is almost perfectly linear (see *Figure 1—figure supplement 1Ci*), which reflects the fact that the quadratic change in CCAF (given by the second derivative of the solution with respect to $t$ in the to the exponential decay model), which has the value 0.01, is tiny compared to the linear change (as are all higher order relationships). Hence, the fact that the parameter estimates from the best-fit exponential decay model produce an almost perfectly linear change in CCAF through the cell cycle, while showing a model fit that is slightly worse than the linear decay model despite having an additional parameter, provides strong support in favour of the linear decay model over an exponential decay model.

To further support this conclusion, we fitted two other generic models for decay of CCAF in the stochastic-deterministic model, a quadratic model of change in CCAF (i.e. where CCAF can change as a function of $t$ and $t^2$) and a cubic model of change in CCAF (i.e. where CCAF can change as a function of $t$, $t^2$ and $t^4$). The quadratic model shows a similar fit to the stochastic-deterministic model with exponential decay in CCAF (adjusted $R$-squared=0.916 and AIC = −54.25 for the quadratic decay model, cf. above). The reason the fits of the quadratic and exponential decay models are so similar is because the best-fit parameters for both models effectively produce linear decay (*Figure 1—figure supplement 1i and ii*), which is reflected in the fact that the estimated quadratic term is near zero (with a value of 0.007). Like the stochastic-deterministic model with exponential decay, the model with quadratic decay is less likely than the model with linear decay because it has an additional parameter (with the ratio of the Akaike weights being 2.88). The cubic model shows a similar fit to the stochastic-deterministic model with linear decay (adjusted $R$-squared=0.921 and AIC = −54.64), but again, because it requires two additional parameters, it is still the less likely model (with the ratio of the Akaike weights being 2.37). Hence, although some degree of non-linearity appears in the best fit parameters for the cubic model (see *Figure 1—figure supplement 1Ciii*), it does not improve the fit of the model, and hence the most likely model is one with linear decay.

## Model fitting with the outlier value included

To confirm that removal of the outlier stalk propensity value measured for *gefE⁻* under G− at 5 hr did not alter our results, we fitted the exponential-decay and stochastic-deterministic models to the full dataset. The exponential-decay model shows the same fit in terms of adjusted $R$-squared as the model with the outlier removed (adjusted $R$-squared=0.88, *AIC* = −50.61), yielding a similar estimated rate of decay ($\lambda$=0.345 compared to 0.351 for the model with the outlier removed) and an identical starting stalk propensity ($\chi$=0.74 for both datasets). Likewise, the stochastic-deterministic model shows approximately the same fit in terms of adjusted $R$-squared as the model with the outlier removed (adjusted $R$-squared=0.91, *AIC* = −57.47), yielding almost identical estimates for starting sensitivity ($C_0^*$=0.557 compared to 0.574 for the model with the outlier removed) and rate of decay in sensitivity ($\beta$=0.397 compared to 0.409 for the model with the outlier removed). The ratio of the Akaike weights of the models with the outlier included is smaller than for the analysis with the outlier removed (30.91 vs 93.46), but still corresponds to a probability of 0.969 that the stochastic-deterministic model is the better fitting model (meaning we can reject the exponential-decay model as the better fitting model with *p*=0.031).

## Influence of stochastic variation on sensitivity to cell-cycle perturbations

To consider how the presence of stochastic variation from noisily expressed genes can buffer against the impact of cell-cycle perturbations on stalk propensity, we developed a simple model for biased sampling across the cell cycle. In the absence of bias, we assume that cells are sampled from a continuous uniform distribution from times zero to one, $U[0, 1]$ so the probability density is the same for all $t$, i.e., $f(t) = 1$ such that $\int_0^1 f(t)\, dt = 1$, meaning that time in the cell cycle is measured as the proportion of the cycle completed (e.g. if $t = \frac{1}{2}$, then cells would be halfway through the cell cycle,

since $\int_0^{0.5} f(t)\,dt = 0.5$). The expected stalk propensity ($P_t$) at time $t$ is given by *Equation (3)*, so the expected propensity of a population of cells sampled uniformly across the cell cycle (denoted $P_u$) is $P_u = \int_0^1 P_t f(t)\,dt$.

Because our primary interest is on the impact of cell-cycle biases, and there is an unlimited range of possible ways for non-random sampling of cells from the cell cycle, we use a simple approach to capture non-random sampling across the cell cycle. We assume that bias is linear and defined by the equation $Q_t = 2\delta t - \delta$, where $Q_t$ gives the relative change in probability density caused by sampling bias, and $\delta$ measures the degree of sampling bias and ranges from $-1$ (bias towards the first half of the cell cycle) and $+1$ (bias towards the second half of the cell cycle). The degree of bias is translated into a bias in probability density function for sampling from the cell cycle as: $N(t) = f(t) + Q_t f(t)$ (see *Figure 8—figure supplement 2* for an illustration of the resulting biased probability density functions). This simple linear modification of the uniform probability density function retains the property that $\int_0^1 N(t)\,dt = 1$ (and hence still represents a probability density function) because it achieves a symmetrically re-distribution of probability density across the range of $t$ from 0 to 1. This makes the expected propensity of a population of cells sampled non-uniformly across the cell cycle (denoted $P_d$): $P_d = \int_0^1 P_t N(t)\,dt$.

There are a number of ways to consider the impact of this bias on the distribution of cells. For simplicity, we consider the proportion of cells that were sampled from the first or second half of the cell cycle, with the proportion of cells in the first half of the cell cycle being $(2 - Q_t)/4$, meaning that, at the maximal degree of bias towards the first half of the cell cycle ($Q_t = -1$), ¾ of cells would be in the first half of the cell cycle (which we denote in *Figure 8—figure supplement 3* as $-¾$). We measured the impact of non-random sampling as the relative change in stalk proportioning: . To consider how stochastic variation impacts the relative change in stalk proportioning, we varied the value of $\sigma$, which gives the standard deviation of the distribution of noisy expression (see *Figure 8—figure supplement 3*).

