## [Editor Report · eLife Assessment]

This **important** study shows how stochastic and deterministic factors are integrated in *Dictyostelium discoideum* to reliably drive determination of distinct cell types despite exposure to nearly identical environmental conditions. The authors present **convincing** evidence that gene expression variability contributes to the robustness of cell fate decisions, which reveals an unexpected role of stochasticity during cell differentiation.

---

## [Referee Report · Reviewer #1 (Public review)]

Summary:

The authors investigate how stochastic and deterministic factors are integrated in cell fate decisions, using **Dictyostelium discoideum** as a model system. They show that cells in different cell cycle phases (a deterministic factor) are predisposed to different fates, albeit with deviations, when exposed to the same environmental stimulus. However, gene expression variability due to asynchrony in cell cycle phase across cells in the populations and stochasticity of biochemical processes enhances the robustness of cellular responses to environmental cues that disrupt the cell cycle.

Using a simple, tractable mathematical model, the authors characterize the response of cell fate decisions as dependent on a combination of deterministic (cell cycle phase) and stochastic factors (variability in gene expression). They then identify Set1 - a key regulator of gene expression variability - and indicate the mechanism of histone methylation, through which it modulates the variability. Finally, they confirm that gene expression variability contributes to the robustness of cells' response (at the population level) by comparing and contrasting the predictions from the mathematical model versus the outcomes in wild type and set1- mutants.

Strengths:

The authors are careful in their choice of experiments and in measuring gene expression variability, using methods that account for expected trends with average gene expression. The mathematical model chosen is simple to follow intuitively and yet predictive enough (at a qualitative level) of the effects of stochastic-deterministic combination of factors, and burst size/frequency.

Weaknesses:

While the authors show that gene expression variation is a feature of genes associated with fate choice and cell type proportioning, it remains somewhat unclear if this kind of variation, or any amount of it, is always beneficial for robustness or there is some optimum level of it.

---

## [Referee Report · Reviewer #2 (Public review)]

Summary:

A fundamental problem in developmental biology is how a group of apparently identical cells breaks symmetry and differentiates into, for instance, type A and type B cells in the absence of any external influence such as a gradient of something causing cells at the left side of the group to become type A cells. The authors use the model system Dictyostelium to explore the interplay between a known cell-cycle-dependent musical chairs mechanism (cells are at random phases of the cell cycle, and a signal that hits all the cells causes cells that happen to be in one set of cell cycle phases to become the A cells, and cells that happen to be in other phases become the B cells), and stochastic gene expression. They identified genes whose expression is stochastic (unusually high cell-cell variation). Using a very clever and elegant genetic screen, they then show that these genes often are associated with cell fate choice. The authors then show that the stochastic genes have reduced levels of histone (H3K4) Me3 methylation, and that a histone methylase called Set1 is important for this process. They then bring the work together to show that the cell-cycle-dependent mechanism and stochastic gene expression work in combination to generate the observed differentiation of Dictyostelium cells.

Strengths:

Combination of theory, clever genetic screens, single-cell RNA-seq, and molecular and cell biology to dive into the fundamental problem of cell fate choice.

Results support the conclusions.

Very significant contribution to developmental biology.

Weaknesses:

Because the paper is co-written by people doing theoretical work and people doing experimental work, the theory sections will be difficult for an experimentalist and vice versa, but it is very much worth the effort to read this paper, there is a lot in here. There are no weaknesses of the methods and results.

---

## [Author Response]

The following is the authors’ response to the original reviews

**Public Reviews:**

**Joint Public Review:**
Summary:The authors investigate how stochastic and deterministic factors are integrated in cell fate decisions, using *Dictyostelium discoideum* as a model system. They show that cells in different cell cycle phases (a deterministic factor) are predisposed to different fates, albeit with deviations, when exposed to the same environmental stimulus. However, gene expression variability (a stochastic factor) enhances the robustness of cellular responses to environmental cues that disrupt the cell cycle.Using a simple, tractable mathematical model, the authors demonstrate that cell fate decisions in *D. discoideum* depend on a combination of deterministic and stochastic factors, i.e., cell cycle phase and gene expression variability, respectively. They then identify Set1 - a key regulator of gene expression variability - indicate the mechanism through which it modulates this variability, and link it to a phenotype in *D. discoideum* development. Finally, they confirm that gene expression variability contributes to the robustness of the cell's response to environmental disruptions that interfere with the cell cycle.Strengths:The authors are careful in the choice of their experiments and in measuring gene expression variability, using methods that account for expected trends with average gene expression.Weaknesses:However, in terms of mathematical modelling, it would be important to rule out sources of stochasticity (other than gene expression variability), and also to consider cases where stochastic factors are not necessarily completely independent of the deterministic ones.

We thank you and the reviewers for the insightful comments that have helped clarify the findings presented. We have addressed all comments and feel that the revised manuscript is much improved.

**Recommendations for the authors:**

**Reviewer #1 (Recommendations for the authors):**
(1) Minor typographical mistakes:(a) in the title: Linage -> lineage

Corrected as suggested

(b) on page 19: use a full stop in "...are biased towards the stalk fate, Use of the cell cycle position..."

Corrected as suggested

(c) on page 20: become -> becoming in "...(and end up biased towards become stalk)..."

Corrected as suggested

(d) on page 16: "mu = G p k". Perhaps it should be x instead of k?

Corrected as suggested

(2) Regarding the abstract:(a) This work tries to outline general principles (coordination/integration of deterministic and stochastic factors) in cell fate choice, especially when cells are faced with (near) identical environmental conditions. Perhaps the abstract, especially the first line, could be rephrased to reflect the generality of symmetry breaking and differentiation that is studied in this article/work. e.g., as was done in the first paragraph of the discussion.

Corrected as suggested

(b) It might be worthwhile clarifying what "this" is in the sentence "We suggest this represents an adaptive mechanism that increases developmental robustness against perturbations that affect deterministic signals." in the abstract.

Corrected as suggested

(3) Regarding the model:(a) The model tries to combine the stochastic and deterministic parts to explain the propensity for stalk fates. It is assumed that the cell cycle-associated factors (CCAF) provide the deterministic part while the cell cycle-independent factors (CCIF) provide the stochastic part. The net result is an addition of the two, which is then compared against a threshold to decide the propensity for stalk fates. However, another simple way to introduce stochasticity would be to make the CCAF decay stochastic. Reasons to consider this scenario would be: (i) the decay process (especially in the biological context) is generally stochastic, (ii) it would not be inconsistent with the fact that cell cycle dependent genes are also variable, and (iii) this way of introducing stochasticity would also provide expression level characteristics/plots similar to the ones outlined in Figure 1C, i.e. with a probability distribution of CCAF values for a given amount of time after mitosis. Would there be arguments or experimental evidence to rule this possibility out? For instance, would the results shown in Figure 7 contradict this model?

We agree that there could be stochasticity the CCAF decay process. In this scenario, the expected value of CCAF (which would reflect the mean of a noisy distribution) would show a deterministic pattern of decay through time, representing the average value of CCAF across cells that are in the same phase of the cell-cycle. The noisiness around such a pattern of deterministic decay in the mean value of CCAF (i.e., the residual variation) would then represent CCIF since it would be, by definition, cell-cycle independent. Hence, the present model is fully consistent with this possibility since it would still lead to some variation being cell-cycle associated and some variation being cell-cycle independent. Therefore, this scenario could be viewed as a different functional/biological process leading to the same ultimate distribution we model. To clarify this, we have added text justifying the hypothesis that the noisy distribution is due to gene expression differences, rather than decay itself:

“Protein levels can vary widely between cells because it is regulated at multiple levels, including transcription, translation and stability. The position of the noisiest step in a pathway affects the overall noise dramatically, because each step usually amplifies noise in the previous steps (Alon 2007). Consistent with this idea, theory and single-cell experiments have shown that a major contributor to cell-cell variation is the bursty expression of low-copy mRNAs. We therefore hypothesized that this noisiness across cells arises from stochastic expression of a set of genes contributing to CCIF levels.”

(b) On page 7, the formula for total CCIF variance assumes independence of the genes g_i. Is this a reasonable assumption?

This concerns the argument that a set of stochastically expressed genes will yield an approximately Gaussian distribution of CCIF. Our results do not depend on the solution for the mean and the variance, only that noisy genes will generally yield such a Gaussian distribution.This is because independence is not strictly required for the central limit theorem to yield a Gaussian distribution. The distribution will still be Gaussian under a broad range of conditions (especially since gene expression is bounded, so there is no chance of the total ending up generating an infinite variance). The primary requirement is that the expression of any given gene is independent from that of most other genes. As a result, most of the variation in expression across genes is independent (even if any given gene is not independent from all other genes).

The most likely pattern of non-independence will be the case in which gene expression is ‘modular’, where there are co-expressed blocks, meaning that non-independence is limited in scale so that genes within a co-regulated block show correlated expression, but their expression is uncorrelated to genes in other blocks. This pattern is functionally analogous to what is known as m-dependence in sequences of random variables (e.g., time series), where variables close together in sequence are correlated (but otherwise uncorrelated). Derivations of the central limit theorem have shown that the means (and hence the sum) of these sorts of variables still follow an approximately Gaussian distribution over a broad range of scenarios. In the case of non-independent gene expression, this means that we can view the independent random variable as being the expression value of a group of co-expressed genes (instead of individual genes). Hence, the means (or sums) of these values will still conform to the central limit theorem.

This problem is addressed in:

Diananda, P. H. 1955. The central limit theorem for m-dependent variables. Proc. Combin. Philos. Soc. 51:92-95

Hoeffding, W. & H. Robbins. 1948. The central limit theorem for dependent random variables. Duke Math. J. 15:773-780

Orey, S. A. 1958. Central limit theorems for m-dependent random variables. Duke Math. J. 25:543-546

Rosén, B. 1967. On the central limit theorem for sums of dependent random variables, Z. Wahrscheinlichkeitstheorie und Verw. Gebiete, 7:48-82

To clarify this, we have added the following text and references:

Although this derivation implicitly assumes that stochastically expressed genes are independent, this assumption is not strictly required for the distribution of CCIF to be approximately normal. If stochastically expressed genes show clustered co-expression owing to shared regulation, then the sum across these co-expressed blocks is still expected to be approximately normally distributed (as long as there are a reasonably large number of co-expressed clusters) (Diananda 1955; Hoeffding and Robbins 1994; Rosén 1967).

(4) In section "Cell cycle independent stochastic gene expression variation is extensive in growing cells":Regarding the statement: "We first determined the coefficient of variation (CV2) of expression for all genes. As expected, this tends to decrease as average expression level increases (Supplementary Figure 2).":It would be good to specify how the "expected variation" was calculated exactly. For instance, it was hard to discern from Supplementary Figure 2 how CV^2 decreasing with average expression levels was used in the calculation of expected variation.

This is described in the methods on page 38

“A trend line was fitted to the data using non-linear least squares regression (Scran v1.15.9). Genes were defined as variable (2073 genes) based on a one-sided test assuming a normal distribution around the trend but one where deviation changed depending on the mean expression of a given gene (Scran v1.15.9 - modelGeneCV2) with a FDR of < 0.05.”

(5) In section "Stochastically expressed genes are associated with cell fate determination"(a) For readers unfamiliar with the organism ‘*Dictyostelium discoideum*’, a short description of its life cycle with growth and development/differentiation phases would be useful to provide the right context.

Corrected as suggested

(b) In section "Cell cycle independent stochastic gene expression variation is extensive in growing cells", it was shown that cell cycle dependent genes are also highly variable (in other words, ‘stochastic’). It would, therefore, be useful to elaborate on the definitions of "stochastically expressed genes, cell cycle-associated genes, and non-variable genes", as used in this section. Admittedly, the distinction does get clearer towards the last section of Results, but some elaboration here would make the reading smoother.

Corrected as suggested

(c) If the "cell cycle associated genes" are the same as "cell cycle dependent genes", it would be good to use one term consistently.

Corrected as suggested

(d) The developmental index is divided into 10 bins from 0 to 1. Is there a rationale for the choice of a number of bins? Would this choice affect significance tests for "stochastic" vs others?(The same question may apply to the "Cell type index")

Significance is robust to the number of bins chosen (e.g. 5-25). Of course, if there are too many bins (low number of genes) or too few bins (addition of noisy data) significance falls. In the case of developmental index, our choice of bins is also based on previous analyses (de Oliveira, et al 2019), which developed the index we used, and showed that a threshold of >0.9 can be used to identify ‘developmentally expressed genes’.

(6) In Figure 5:(a) Does the statement "*** binomial test, p<0.01." (as seen in caption for part C) actually refer to part D?

Corrected as suggested

(b) Could the authors please specify what "mis-expressed" means in Figure 5D? Are these genes that are upregulated, downregulated, or both? From what set of genes was the random sampling done?

Corrected as suggested

(c) In Figure 5F, is the decrease in CV^2 explained entirely by the increase in mean (as shown in Figure 5E)?

We appreciate the point made by the reviewer and recognise that disentangling changes in gene expression variation from changes in expression levels is extremely difficult (any changes in burst frequency will necessarily affect expression level). However, we do not think this affects our conclusions, which are supported by results with representative Set1 dependent reporter genes (Figure 5G and H) which suggest that the number of cells expressing (rather than the expression in each cell is affected) in these cases at least.

(7) In Figure 6A: Could the authors please elaborate on the difference between the rows labelled "WT" and "set1-"? Are they two different types of chimera?

Corrected as suggested

(8) In Section "Cell cycle position and gene expression variation interact to control cell type proportioning":Is there a graph corresponding to the statement "However, the level of GFP expression in each responding cell did not significantly change."?

Corrected as suggested

(9) In section "Influence of stochastic variation on sensitivity to cell cycle perturbations" of the Supplementary text:(a) The model for cell cycle bias is not entirely clear. For instance, is the quantity N(t) = U(t) + Q_t U(t) also a probability distribution, like U(t) is? If so, there must be a normalization factor. It was difficult to understand the procedure behind this calculation. Perhaps some more elaboration (with words or a small schematic) on this model/method would help.

The value of U(t) was originally being used to denote the uniform probability density function (for the uniform distribution), but for clarity this has been changed to follow the convention that U[a,b] denotes the uniform distribution over the interval from a to b (which, in this case would be U[0, 1]), while f(t) is now being used to make it clear that this is the probability density, where f(t) = 1 across the interval. Because the uniform distribution necessarily integrates to 1 over the defined range, it does not need to be normalised. The confusion here is perhaps due to the expression f(t) = 1 being interpreted as defining the probability of sampling a value of t (but in a continuous distribution we can only define the probabilities of sampling over an interval), instead of defining the probability density over the interval from a to b, where f(x) would be 1/(b – a), and hence over the interval of 0 to 1, f(x) would equal 1.

To help clarify this issue, this section has been rewritten and a new figure (which appears as Supplementary Figure 12) has been added that illustrates the resulting probability density functions for biased sampling from the cell cycle.

(b) References to Figure 8A, B seem to be indicating Supplementary Figure 12 instead.

Corrected as suggested

**Reviewer #2 (Recommendations for the authors):**
This manuscript seems quite interesting, but many sections are so unclear that I cannot follow what has been done. I would suggest slowly going through the manuscript and carefully explaining things. This will probably considerably increase the size of the manuscript, but many sections are too terse to follow even after many, many readings of the Results and figure legend.

Corrected as suggested

Some specific comments (this is not at all comprehensive, but rather illustrative)

Page 2 - 'genes strongly associated with fate choice' - can you explain this a bit more - genes associated with one cell type or another, or genes that somehow regulate the choice?

Corrected as suggested

Page 2 - this abstract is quite vague, I would suggest being more specific to reflect what is in the manuscript.

Corrected as suggested

Page 3 - 'exhibit bivalent H3K4me3..' please explain 'bivalent' a bit more.

Corrected as suggested

Page 7 - 'Bernoulli process with probability that (meaning that is scaled to the size of the temporal interval)' (non-copying symbols deleted) could be simplified.

Corrected as suggested

Page 7 - please define all variables/ equation components. What is N? What is x bar? What is s2? The middle paragraph is very difficult to follow.

This paragraph has been rewritten and a definition of the distribution added for clarity.

Page 7 - 'genes might logically vary in the value of pi, such variability does not impact our results. Trying to decipher this paragraph, it seems that pi is a function of time, so this could affect the results.

pi is the probability that a stochastically expressed gene is actually expressed in whatever interval is being considered for all genes. pi will necessarily increase if the time interval considered is increased. The key point is we are considering the probability that any given gene is expressed in the same time interval. In this case, genes could vary in pi, and thus some burst more often and others less often.

Page 9 - '(it is 98.35 times more likely' there may be too many significant figures here).

Corrected as suggested

Page 10 - for the Area Under the Receiver Operating Characteristic Curve (AUROC), what are you classifying? AUROC is typically used for diagnostic tests to determine how well the test can discriminate between two completely different outcomes. What is the input, and what are the outcomes?

Corrected as suggested

Figures:What are the dashed lines in Figure S2A?

Corrected as suggested

What are the X-axes in Figure S3?

Corrected as suggested

I do not understand what you are showing in Figure S3.

Corrected as suggested in results

In Figure 2B, I cannot find in the text or figure legend any description or explanation of 'Group 1', 'Group 2', or 'Group 3'.

Corrected as suggested

Figure 3D needs a lot more explanation; I cannot understand this based on the text and the figure legend.

Corrected as suggested

The Set1 work should discuss the work in PMID: 39242621

Corrected as suggested

Figure 8 D needs a size bar

Corrected as suggested